# Managing disease outbreaks: The importance of vector mobility and spatially heterogeneous control

**Jeffery Demers**[1]*, **Sharon Bewick**[2], **Folashade Agusto**[3], **Kevin A. Caillouët**[4], **William F. Fagan**[1], **Suzanne L. Robertson**[5]

**1** Department of Biology, University of Maryland College Park, College Park, Maryland, United States of America, **2** Department of Biological Sciences, Clemson University, Clemson, South Carolina, United States of America, **3** Department of Ecology and Evolutionary Biology, University of Kansas, Lawrence, Kansas, United States of America, **4** St. Tammany Parish Mosquito Abatement District, Slidell, Louisiana, United States of America, **5** Department of Mathematics and Applied Mathematics, Virginia Commonwealth University, Richmond, Virginia, United States of America

* jdemers@umd.edu

## Abstract

Management strategies for control of vector-borne diseases, for example Zika or dengue, include using larvicide and/or adulticide, either through large-scale application by truck or plane or through door-to-door efforts that require obtaining permission to access private property and spray yards. The efficacy of the latter strategy is highly dependent on the compliance of local residents. Here we develop a model for vector-borne disease transmission between mosquitoes and humans in a neighborhood setting, considering a network of houses connected via nearest-neighbor mosquito movement. We incorporate large-scale application of adulticide via aerial spraying through a uniform increase in vector death rates in all sites, and door-to-door application of larval source reduction and adulticide through a decrease in vector emergence rates and an increase in vector death rates in compliant sites only, where control efficacies are directly connected to real-world experimentally measurable control parameters, application frequencies, and control costs. To develop mechanistic insight into the influence of vector motion and compliance clustering on disease controllability, we determine the basic reproduction number $R_0$ for the system, provide analytic results for the extreme cases of no mosquito movement, infinite hopping rates, and utilize degenerate perturbation theory for the case of slow but non-zero hopping rates. We then determine the application frequencies required for each strategy (alone and combined) in order to reduce $R_0$ to unity, along with the associated costs. Cost-optimal strategies are found to depend strongly on mosquito hopping rates, levels of door-to-door compliance, and spatial clustering of compliant houses, and can include aerial spray alone, door-to-door treatment alone, or a combination of both. The optimization scheme developed here provides a flexible tool for disease management planners which translates modeling results into actionable control advice adaptable to system-specific details.

**Data Availability Statement:** All relevant data are within the manuscript and its Supporting Information files.

**Funding:** This work was supported by a grant from the Simons Foundation (426126, SR), a University of Kansas General Research Grant (2301-2105075, FBA), and a Department of Defense SERDP contract (W912HQ-16-C-0054, SB and WFF). The funders had no role in study design, data collection and analysis, decision to publish, or preparation of the manuscript.

## Author summary

Mosquitoes spread diseases, including West Nile virus, dengue, and most recently, Zika. Control of mosquitoes in residential areas involves a combination of aerial spraying from planes and door-to-door treatment of individual yards. There are benefits to both approaches. With aerial spraying, it's easy to reach every yard; however, aerial pesticides are short-lived and less effective. With door-to-door treatment, pesticides are longer lasting and more effective; however, not all people allow officials into their yards. Consequently, large portions of a neighborhood can go untreated, leaving ample mosquito habitat. We study how the optimal combination of aerial spraying and door-to-door treatment varies with the fraction of houses that allow treatment, as well as with clustering of non-compliant houses, and the extent to which mosquitoes move from one yard to another. Overall, we find that aerial spraying is best at low compliance levels and when non-compliant houses are clustered. At high compliance levels and when non-compliant houses are dispersed, door-to-door treatment is most cost-effective. Finally, there are intermediate scenarios where combinations of aerial spraying and door-to-door treatment are optimal. Interestingly, less mobile mosquitoes are harder to control, because they can 'hide' in inaccessible habitats. This allows diseases to spread in these localized regions.

## Introduction

The increased worldwide emergence and re-emergence of vector-borne diseases seen in recent decades demands increasingly efficacious responses from health and government agencies for the prevention of public health crises [1–3]. Zika, for example, a flavivirus spread primarily between humans and mosquitoes with links to Gullain-Barré syndrome [4] and congenital brain defects [5], remained sequestered to the Eastern hemisphere until 2013 [6]. The first evidence of local Zika transmission on the United States mainland was reported by the Centers for Disease Control and Prevention (CDC) on July 29, 2016, when four cases of human infection were confirmed in Miami-Dade County, Florida [7]. Days later, the CDC subsequently labeled a 1 square mile block of the Wynwood neighborhood (located within the city of Miami) an active Zika transmission zone [8], followed by 4.5 square miles of Miami Beach and 1 square mile of the Little River neighborhood in October [9]. When an active transmission zone is identified, the CDC recommends focusing and intensifying vector control efforts within the immediate vicinity of the area in order to keep diseases localized [10]. Due in part to intensified control efforts, by the end of 2016, all zones in Miami-Dade County had been cleared of active transmission status after going 45 days with no detected cases of new local transmission [11, 12]. The Miami-Dade County outbreak suggests that Zika spread in human population centers is highly localized, with 256 cases reported over a small number of neighborhood areas no more than a few square miles in size [13, 14].

In Miami-Dade County and elsewhere, officials control and manage localized vector-borne disease outbreaks like the 2016 Zika event through integrated vector management strategies [2, 3, 15]. Strategies typically consist of regular, large-scale larvicide and adulticide application by airplanes and/or street trucks, combined with door-to-door control efforts that require officials be given access to yards and other private property where they check for and eliminate sources of standing water susceptible to oviposition, in addition to applying larvicide and adulticide precisely and thoroughly. Both strategies have advantages and disadvantages. Aerial spraying can quickly provide blanket coverage to an entire neighborhood area. However, adulticide aerial spray is only effective against active mosquitoes who come into contact with

short-lived airborne insecticide plumes, so repeated frequent applications may be required to achieve sufficient levels of control [15]. Repeated application of aerial adulticide can become costly and promote insecticide resistance, and residents may be uncomfortable with planes flying overhead frequently [3, 16]. Therefore, aerial spraying alone may not always be a viable strategy. With door-to-door control, a longer lasting, more effective residual barrier adulticide spray can be applied to vegetation and other mosquito landing surfaces in individual yards, and potential larval habitats (e.g. receptacles for standing water) can be identified and removed. This approach, however, is reported to be costly and time-consuming, and residents may decline to admit control officials access to their yard or may not be at home when control is being implemented, thereby making large-scale neighborhood coverage and sufficient levels of control potentially difficult to achieve [15]. The inaccessible yards within a neighborhood may be randomly dispersed or highly clustered, potentially due to social influences or other interactions between individual neighbors, and the clustering of control access can influence overall control efficacy [17]. Further, although accessible and inaccessible yards are spatially localized to individual sites or clusters of sites, mosquito motion allows localized heterogeneous levels of vector control to produce effects over larger, potentially neighborhood-wide, spatial scales [17, 18].

Aerial spraying and door-to-door control strategies provide a trade-off between small-scale localized control efficacy and ease in achieving efficient large-scale neighborhood-wide control coverage [15]. This trade-off, together with local social, political, and economic concerns, makes the design and implementation of effective integrated vector management strategies a logistical challenge [3, 16]. Given the complexities inherent to designing integrated vector management programs, mathematical models are useful tools for analyzing and predicting the efficacy of intervention strategies. In this paper, we develop an ordinary differential equation (ODE) model to analyze the efficacy of integrated vector management strategies, specifically those comprised of adulticide aerial spraying and door-to-door control, for preventing vector-borne disease outbreaks in at-risk neighborhood-scale environments comparable to the 2016 Miami-Dade Zika active transmission zones.

Many previous studies have analyzed integrated management strategies using ODE models assuming a single-patch homogeneous system of hosts and vectors. Such models have analyzed control efficacy for reducing mosquito populations [19, 20], and also for mitigating specific vector-borne diseases like dengue [21–23], West Nile virus [24, 25], malaria [26], and Zika [27]. Many models incorporate control in a simplistic but mathematically tractable manner through its overall gross effects on model parameters, often assuming time-independent controls and analyzing intervention strategies through the model basic reproduction number [20, 24, 26, 27]. Models of this class have also successfully applied optimal control theory to derive time-dependent control protocols which optimally balance measures of control cost against measures of disease severity [20, 23, 25, 26]. Other models consider impulsive control protocols with finite natural efficacy times which are mathematically more complicated, but are also more realistic and more directly connected to real-world actionable control advice [19, 22]. A few studies explore both approaches to control modeling [21, 25]. In contrast, this paper utilizes a hybrid approach to control modeling introduced in Ref. [28]. Specifically, we incorporate control through its overall gross effects on model parameters, where the degrees of control's influence are given as functions of real-world experimentally measurable control parameters, costs, and impulse application frequencies, and we quantify intervention efficacy through the basic reproduction number to determine actionable control advice in the form of cost-optimal impulse application schedules.

Single-patch models, while useful for studying control strategies and disease dynamics, are unable to capture features paramount to neighborhood-scale management strategies,

specifically heterogeneous control, control clustering, and vector motion. Metapopulation models incorporate individual homogeneous patches of vector and host populations connected via a network structure allowing host and/or vector movement between connected patches, and are natural candidates for modelling disease systems with spatial heterogeneities at both large and small spatial scales. Like single-patch models, metapopulation models study control strategies for reducing vector populations [17], as well as control strategies for mitigating specific vector-borne diseases, where control efficacy is determined using the basic reproduction number [29–31], optimal control theory [32, 33], or impulsive controls with naturally decaying efficacies [34–36]. Results from metapopulation models have shown that network connectivity allows for disease levels to persist in patches where they would otherwise face rapid extinction, which can have important implications for designing vector management strategies [31, 37, 38]. Many metapopulation studies are concerned primarily with movement between patches which have no a priori spatial relationship to one another, so the models of motion used in such studies are not entirely mechanistic. In this paper, our goal is to provide actionable control advice which could in principle be adapted to experimentally measurable conditions encountered in the field, so we prefer a mechanistic model of motion. In this regard, our model is similar to the metapopulation model of Lutambi et al. in Ref. [17], which incorporates heterogeneous integrated vector management strategies and vector motion under various levels of control clustering. However, the model in Ref. [17] determines control efficacy through vector population reduction with no explicit regard to disease dynamics, while we are concerned with control efficacy for disease suppression as determined by the basic reproduction number.

To capture the complexities of vector management on neighborhood spatial scales with heterogeneous control access and mobile mosquitoes, in the paper, we construct a metapopulation model consisting of a grid of square patches representing residential properties connected via nearest neighbor mosquito motion. Within the grid, we assume that aerial spray control covers the entire neighborhood, while access to properties by door-to-door control workers varies spatially. To analyze the efficacy of integrated vector management strategies, we first study the model's basic reproduction number $\mathcal{R}_0$, and we determine to what extent $\mathcal{R}_0$ is influenced by vector motion and the strength of control in accessible and inaccessible sites. The goal of this analysis is to provide mechanistic insight into the effects of spatially heterogeneous control and vector motion on epidemic outbreak potential. In this effort, we derive expressions for $\mathcal{R}_0$ in simplified but analytically tractable special cases, and we utilize degenerate perturbation theory, a technique most commonly applied in physics to determine perturbations to quantum energy levels, to analyze the case of slow mosquito motion. Next, for each type of control (aerial and door-to-door), we associate the degree of the control's influence on model parameters with real-world experimentally measurable control parameters and costs, and we determine the application frequency needed to reduce the basic reproduction number to one, as well as the optimal monetary costs associated with doing so. In this analysis, we assess under what conditions door-to-door control alone can suppress an outbreak, and when it can do so at less cost than with aerial spray alone. We also assess under what conditions cost-optimal strategies call for door-to-door control alone, aerial spray control alone, or a combined door-to-door/aerial spray strategy. We investigate how the answers to these questions depend on the percentage of compliant houses that admit officials into their yard, the physical location of compliant/non-compliant houses, and the speed with which mosquitoes move between yards. The cost-optimization scheme we develop for this analysis is mathematically simple in comparison to the Pontryagin maximum principle and Bellman equation techniques typically utilized throughout the optimal disease control literature [39]. Despite its simplicity, however, our scheme yields actionable control advice for real-world intervention methods. The model

and optimization scheme presented in this paper are straightforward to adapt to system-specific disease parameters, as well as to natural and control-related spatial heterogeneities which may be encountered in the field, and therefore may serve as a useful tool for disease-management professionals seeking to better plan vector control programs.

## Methods

### Model development

We model a residential neighborhood as a two-dimensional lattice of $\mathcal{N} \times \mathcal{N}$ square patches indexed by integers $i$, where $i \in \{1, 2, \ldots, \mathcal{N}^2 - 1, \mathcal{N}^2\}$, laid out as in Fig 1. Each patch represents a physical area of size $l \times l$, and we assume $N_i^h$ humans reside in a home located within patch $i$. A typical patch encompasses the dwelling, the surrounding property, and a portion of the adjacent street. We posit that while present in the neighborhood, humans spend the majority of their time in and around their home patch, and that the time spent traveling throughout the neighborhood is negligible by comparison. This description of human activity is applicable, for example, to residents of neighborhoods who typically use a vehicle or public transportation when they need to leave the area (as opposed to walking or biking through the neighborhood). Mosquitoes, on the other hand, move continually in a somewhat random manner, and they may pass through many patches over the course of a lifetime depending on the physical patch size; *Aedes aegypti*, for example, will disperse hundreds to thousands of meters from their emergence site over the course of a lifetime [40]. Based on this observation, we consider a simplified model in which mosquitoes are able to move between neighboring patches while humans remain fixed at their home patch.

| 1 | 2 | 3 | ..... | $\mathcal{N}$-1 | $\mathcal{N}$ |
|---|---|---|---|---|---|
| $\mathcal{N}$+1 | $\mathcal{N}$+2 | $\mathcal{N}$+3 | ..... | $2\mathcal{N}$-1 | $2\mathcal{N}$ |
| $2\mathcal{N}$+1 | $2\mathcal{N}$+2 | $2\mathcal{N}$+3 | ..... | $3\mathcal{N}$-1 | $3\mathcal{N}$ |
| ⋮ | ⋮ | ⋮ | | ⋮ | ⋮ |
| $(\mathcal{N}$-2$)\,\mathcal{N}$+1 | $(\mathcal{N}$-2$)\,\mathcal{N}$+2 | $(\mathcal{N}$-2$)\,\mathcal{N}$+3 | ..... | $(\mathcal{N}$-1$)\,\mathcal{N}$-1 | $(\mathcal{N}$-1$)\,\mathcal{N}$ |
| $(\mathcal{N}$-1$)\,\mathcal{N}$+1 | $(\mathcal{N}$-1$)\,\mathcal{N}$+2 | $(\mathcal{N}$-1$)\,\mathcal{N}$+3 | ..... | $\mathcal{N}^2$-1 | $\mathcal{N}^2$ |

**Fig 1. Site indices and corresponding locations within the neighborhood.**

**Mosquito motion.** Let $N_i^\nu(t)$ denote the distribution of mosquitoes evaluated at patch $i$ at a time $t$. We model mosquito motion with a nearest neighbor random walk between patches, and we assume the following population dynamics for $N_i^\nu(t)$:

$$\dot{N}_i^\nu(t) \quad = \quad \Lambda_i - \mu_i N_i^\nu + \sum_{j=1}^{\mathcal{N}^2} \omega_{ij} N_j^\nu(t), \tag{1}$$

where the overdot denotes a time derivative, $\Lambda_i$ denotes the mosquito emergence rate in patch $i$, $\mu_i$ denotes the per-capita mosquito death rate in patch $i$, $\omega_{ij}$ denotes the transition probability per unit time for an individual mosquito to hop from patch $j$ to patch $i$ for $j \neq i$, and $\omega_{ii}$ is defined as $-1$ multiplied by the probability per unit time for a mosquito to transition out of patch $i$:

$$\omega_{ii} \quad = \quad -\sum_{\substack{j=1 \\ j \neq i}}^{\mathcal{N}^2} \omega_{ji}. \tag{2}$$

The parameters $\Lambda_i$ and $\mu_i$ will vary across the network due to mosquito control efforts. Assuming unbiased nearest neighbor hopping with hopping rate $\omega$, we can write $\omega_{ij}$ as

$$\omega_{ij} = -\omega \mathcal{L}_{ij}, \tag{3}$$

where $\mathcal{L}_{ij}$ denotes the elements of an $\mathcal{N}^2 \times \mathcal{N}^2$ Laplacian matrix associated with nearest neighbor connectivity of our network. The actual form of the Laplacian matrix will depend on the boundary conditions we choose for the edge of our neighborhood. We desire a mathematically closed system which prohibits mosquitoes from fluxing in or out of the neighborhood boundary, so we must choose between periodic or reflecting boundary conditions. The choice is immaterial for the mathematics developed throughout our Methods, but will be needed for some numerical simulations in our Results. For numerical simulations which consider specific neighborhood configurations, we compare results for both sets of boundary conditions (our results will show that the choice makes little practical difference). For numerical simulations which require large numbers of neighborhood configurations to be generated, we choose periodic boundary conditions for simplicity. The form of both Laplacian matrices are given in S1 Appendices Sec. 1 for reference.

Throughout this paper, we leave the hopping rate as a variable parameter and avoid focusing on specific numerical values wherever possible. The hopping rate may vary widely between real systems based on the vector species, environmental conditions, and the physical patch size, so in order to apply the results of this paper to the field, one must find an estimate for the hopping rate based on the specifics of the real system under consideration. In this regard, it is helpful to recognize that the discrete patch/hopping system is a simplified approximation for a more realistic continuous diffusion system. For a two-dimensional diffusion process with diffusion constant $D$, the hopping rate under the discrete approximation scales with $D$ according to $D \approx \omega l^2$, where $l^2$ is the physical patch size. For example, using the diffusion constants from previous modeling studies for *Aedes albopictus* in Ref. [41] and *Aedes aegypti* in Refs. [18] and [42], we obtain the values $\omega = 0.288$ day$^{-1}$, $\omega = 0.09$ day$^{-1}$, and $\omega = 25.0$ day$^{-1}$, respectively, assuming a patch size $l^2 = 500$ m$^2$. The diffusion constant can in principle be determined experimentally by releasing a large number of mosquitoes into the system under consideration at a single point, and then measuring their mean squared displacement $\langle \Delta \mathbf{x}^2 \rangle$ over their natural lifetime $1/\mu_0$. In two-dimensions, the mean squared displacement over a time $1/\mu_0$ is related to the diffusion constant via $\langle \Delta \mathbf{x}^2 \rangle = 4D/\mu_0$.

**SEIR model equations.** We consider the following SEIR model for vector-borne disease spread throughout the neighborhood:

$$\dot{S}_i^v(t) \;=\; \Lambda_i - \mu_i S_i^v(t) - \omega \sum_{j=1}^{\mathcal{N}^2} \mathcal{L}_{ij} S_j^v(t) - \beta^v b \frac{I_i^h(t)}{N^h} S_i^v(t) \tag{4a}$$

$$\dot{E}_i^v(t) \;=\; -\mu_i E_i^v(t) - \omega \sum_{j=1}^{\mathcal{N}^2} \mathcal{L}_{ij} E_j^v(t) + \beta^v b \frac{I_i^h(t)}{N^h} S_i^v(t) - p^v E_i^v(t) \tag{4b}$$

$$\dot{I}_i^v(t) \;=\; -\mu_i I_i^v(t) - \omega \sum_{j=1}^{\mathcal{N}^2} \mathcal{L}_{ij} I_j^v(t) + p^v E_i^v(t) \tag{4c}$$

$$\dot{S}_i^h(t) \;=\; -\beta^h b \frac{S_i^h(t)}{N^h} I_i^v(t) \tag{4d}$$

$$\dot{E}_i^h(t) \;=\; \beta^h b \frac{S_i^h(t)}{N^h} I_i^v(t) - p^h E_i^h(t) \tag{4e}$$

$$\dot{I}_i^h(t) \;=\; p^h E_i^h(t) - r I_i^h(t) \tag{4f}$$

$$\dot{R}_i^h(t) \;=\; r I_i^h(t), \tag{4g}$$

where $S_i^v(t)$, $E_i^v(t)$, and $I_i^v(t)$ denote the distribution of susceptible, exposed, and infectious mosquitoes in patch $i$ at time $t$, respectively, and $S_i^h(t)$, $E_i^h(t)$, $I_i^h(t)$ and $R_i^h(t)$ denote the distribution of susceptible, exposed, infectious, and recovered hosts in patch $i$ at time $t$. As we are primarily interested in the heterogeneities introduced by compliance to control efforts, we assume the total number of hosts in each patch is a fixed constant $N^h$ (the $N^h$ hosts in patch $i$ represent the humans who live in the residential dwelling located in patch $i$). Similarly, the per-bite human-to-vector and vector-to-human transmission probabilities, $\beta^v$ and $\beta^h$, respectively, the extrinsic and intrinsic incubation periods, $1/p^v$ and $1/p^h$, respectively, and the average human recovery time, $1/r$, are all assumed to be patch-independent. We assume human birth and death rates are negligible over the time scale of interest and that all infectious mosquitoes die before recovering. The meaning of each parameter, as well as the example values used for numerical simulations, is summarized in Table 1.

**Control and compliance.** We incorporate door-to-door and adulticide aerial spray control strategies into our neighborhood disease model as patch-dependent, time-independent reductions and increases in vector emergence and death rates, respectively. Controls incorporated in this manner are simple but effective models for describing the effects of regularly repeated fixed strategy real-world controls (e.g. an aerial spraying campaign in which airplanes deploy a fixed amount of pesticide repeatedly according to a fixed schedule, or a door-to-door control campaign in which employees visit residences repeatedly according to a fixed schedule) [28]. Under this modeling methodology, the optimal control protocols we find for reducing the basic reproduction number will be naturally phrased in terms of actionable control advice based on real-world control parameters.

In the absence of control, we assume homogeneous natural death and emergence rates, denoted by $\mu_0$ and $\Lambda_0$, respectively, in each patch throughout the neighborhood. Adulticide aerial spray is an area-wide control assumed to cover the entire neighborhood equally, so we

**Table 1. Model parameters and values used in numerical simulations.** The values here are taken to represent a typical sample of *Ae. aegypti* population parameters and a typical sample of disease parameters for a vector-borne disease such as Zika in North America (see Ref. [27] and the references contained within). These parameters are expected to vary widely between locations, mosquito species, and diseases, and values should be carefully estimated when applying the techniques in this paper to specific situations in the field.

| Parameter | Description | Value |
|---|---|---|
| $\mathcal{N}^2$ | Number of patches | 100 |
| $N^h$ | Humans per patch | 4 |
| $\Lambda^0$ | Natural mosquito emergence rate per patch | 3 day$^{-1}$ |
| $\mu^0$ | Natural mosquito death rate | 1/14 day$^{-1}$ |
| $b$ | Mosquito biting rate | 1 / 3 day$^{-1}$ |
| $1/p^v$ | Extrinsic incubation period | 14 days |
| $1/p^h$ | Intrinsic incubation period | Irrelevant |
| $1/r$ | Average human recovery time | 7 days |
| $\beta^v$ | Human to mosquito transmission probability | 0.25 |
| $\beta^h$ | Mosquito to human transmission probability | 0.25 |
| $\omega$ | Mosquito hopping rate | Variable |
| $\Lambda_N$ | Controlled non-compliant patch mosquito emergence rate | 3 day$^{-1}$ |
| $\Lambda_C$ | Controlled compliant patch mosquito emergence rate | Variable |
| $\mu_N$ | Controlled non-compliant patch mosquito death rate | Variable |
| $\mu_C$ | Controlled compliant patch mosquito death rate | Variable |

model this strategy as a uniform increase in vector death rates at every site. Door-to-door control affects only the 'compliant' sites for which the residents permit yard access to government or health agency workers to apply residual barrier adulticide spray and conduct larval habitat reduction. We model this strategy as uniform increase in vector death rate and decrease in vector emergence rate in compliant sites only. We thus have three possible assignments for the vector death rate in patch *i*:

$$\mu_i = \begin{cases} \mu_0, & \text{No controls applied to system} \\ \mu_N, & \text{Controlled system, patch } i \text{ non-compliant} \\ \mu_C, & \text{Controlled system, patch } i \text{ compliant} \end{cases} \qquad (5)$$

where $\mu_0 \leq \mu_N \leq \mu_C$. Likewise, the vector emergence rate in patch *i* can take one of three values:

$$\Lambda_i = \begin{cases} \Lambda_0, & \text{No controls applied to system} \\ \Lambda_N, & \text{Controlled system, patch } i \text{ non-compliant} \\ \Lambda_C, & \text{Controlled system, patch } i \text{ compliant} \end{cases} \qquad (6)$$

where $\Lambda_0 \geq \Lambda_N \geq \Lambda_C$. Our model does not contain a larval class, so larvicide controls for both area-wide spraying and door-to-door strategies applied in conjunction with larval source reduction controls are outside the scope of our work (see Ref. [28] for a detailed explanation), and are not considered in this paper. We will therefore always have $\Lambda_0 = \Lambda_N$, but we retain the above notation to emphasize the structure of compliant and non-compliant sites.

### The basic reproduction number

We evaluate the efficacy of control strategies for suppressing epidemic outbreaks through the model basic reproduction number $\mathcal{R}_0$. The basic reproduction number is a general measure of outbreak severity which, for our system, determines the stability of the disease-free equilibrium against small perturbations in the infected population [43]: perturbing an entirely susceptible population will result in epidemic outbreak if $\mathcal{R}_0 > 1$ and disease die-out if $\mathcal{R}_0 < 1$. To calculate the basic reproduction number as a function of model parameters, we utilize the next generation matrix method as outlined in [44]. Under this formalism, $\mathcal{R}_0$ provides a measure of the maximum number of infected individuals generated over the lifetime of a single infected individual introduced into a background system held at 100% susceptible disease free equilibrium. In S1 Appendices Sec. 3, we show that $\mathcal{R}_0$ can be found by finding the largest non-negative eigenvalue solution to either of the following eigenvalue problems:

$$\mathcal{R}_0^2 \mathbf{I}^h = \mathcal{M} \mathbf{I}^h \tag{7}$$

$$\mathcal{R}_0^2 \mathbf{I}^v = \underline{\mathcal{M}} \mathbf{I}^v. \tag{8}$$

We refer to the $\mathcal{N}^2 \times \mathcal{N}^2$ matrices $\mathcal{M}$ and $\underline{\mathcal{M}}$ as the "second generation matrices." The $i, j$ component of $\mathcal{M}$ represents the expected number of infectious hosts generated in site $i$ over the lifetime of a single infectious host introduced in site $j$ who passes the disease through the vector population (assuming disease dynamics which are linearized about disease-free equilibrium), and vice versa for infectious vectors and the components of $\underline{\mathcal{M}}$. The second generation matrices are non-negative such that $\mathcal{R}_0^2$ is their largest eigenvalue [45], so by the Perron-Frobenius theorem for non-negative matrices [46], the eigenvectors $\mathbf{I}^h$ and $\mathbf{I}^v$ can be taken to be non-negative such that their components sum to unity. Under this convention, $\mathbf{I}^h$ and $\mathbf{I}^v$ represent the worst case scenario spatial distributions of a given number of initial infectious hosts and vectors, respectively, which produce the largest asymptotic infectious growth rate under the disease dynamics linearized about the disease-free equilibrium. In order to provide biological interpretation to our mathematical results, we determine both $\mathcal{R}_0$ and the corresponding eigenvectors $\mathbf{I}^h$ and $\mathbf{I}^v$. Generally, Eqs (7) and (8) can only be solved numerically. There are, however, a few biologically relevant simplified special cases, detailed in the following subsections, for which one can derive analytic expressions. These cases represent extreme limits of hopping rate and spatial homogeneity which place bounds on more realistic intermediate cases, and will be useful for interpreting numerical results from the optimized control mathematics to be developed. The details of these $\mathcal{R}_0$ derivations are provided in S1 Appendices Sec. 4.

**Isolated sites.**   For a neighborhood comprised of a single isolated site (e.g. $\mathcal{N}^2 = 1$), the hopping rate $\omega$ and Laplacian matrix $\mathcal{L}$ are irrelevant, and our system reduces to a basic single-patch SEIR model. The single site basic reproduction number takes one of the following expressions:

$$
\begin{aligned}
\mathcal{R}_{00} &= \sqrt{\beta^v \frac{b}{r} \beta^h \frac{b}{\mu_0} \frac{p_v}{p_v + \mu_0} \frac{1}{N^h} \frac{\Lambda_0}{\mu_0}} \\
\mathcal{R}_{0N} &= \sqrt{\beta^v \frac{b}{r} \beta^h \frac{b}{\mu_N} \frac{p_v}{p_v + \mu_N} \frac{1}{N^h} \frac{\Lambda_N}{\mu_N}} \\
\mathcal{R}_{0C} &= \sqrt{\beta^v \frac{b}{r} \beta^h \frac{b}{\mu_C} \frac{p_v}{p_v + \mu_C} \frac{1}{N^h} \frac{\Lambda_C}{\mu_C}},
\end{aligned}
\tag{9}
$$

where $\mathcal{R}_{00}$, $\mathcal{R}_{0N}$, and $\mathcal{R}_{0C}$ refer to uncontrolled, non-compliant, and compliant patches, respectively.

**Homogeneous systems.** For the special cases of no control, 100% compliance, and 100% non-compliance, all model parameters are equivalent at every site in the neighborhood, and we find

$$\mathcal{R}_0 = \begin{cases} \mathcal{R}_{00}, & \text{Uncontrolled system} \\ \mathcal{R}_{0N}, & 100\% \text{ non-compliant system} \\ \mathcal{R}_{0C}, & 100\% \text{ compliant system.} \end{cases} \tag{10}$$

The corresponding eigenvectors $\mathbf{I}^h$ and $\mathbf{I}^v$, regardless of control or compliance, are found to be

$$\begin{aligned} \mathbf{I}^h &= \mathbf{I}^v \\ &= \frac{1}{\mathcal{N}^2}\mathbf{1}, \end{aligned} \tag{11}$$

where $\mathbf{1}$ denotes the $\mathcal{N}^2$ dimensional vector comprised of all ones. In other words, $\mathbf{I}^h$ and $\mathbf{I}^v$ are uniformly distributed over the entire neighborhood.

**Infinitely fast hopping.** For the case of infinitely rapid mosquito hopping, we consider the limit $\omega \to \infty$. This is a singular limit, and we utilize the methods of Tien et al. outlined in [47] to perform our calculations (see S1 Appendices Sec. 4.3). We find

$$\mathcal{R}_0 \underset{\omega \to \infty}{=} \sqrt{\beta^v \frac{b}{r} \beta^h \frac{b}{\langle\mu\rangle} \frac{p^v}{p^v + \langle\mu\rangle} \frac{1}{N^h} \frac{\langle\Lambda\rangle}{\langle\mu\rangle}}, \tag{12}$$

where the expression $\langle g \rangle$ denotes the average of a site dependent quantity $g$ over the entire neighborhood. The corresponding eigenvectors are found to be uniformly distributed:

$$\begin{aligned} \mathbf{I}^h &\underset{\omega \to \infty}{=} \mathbf{I}^v \\ &\underset{\omega \to \infty}{=} \frac{1}{\mathcal{N}^2}\mathbf{1}. \end{aligned} \tag{13}$$

**Infinitely slow hopping.** In the infinitely slow hopping $\omega = 0$ limit, all patches decouple from one another. Here and throughout the rest of this paper, we use the subscript (0) to refer to quantities evaluated for the special case $\omega = 0$. The basic reproduction number $\mathcal{R}_{0(0)}$ is found to be

$$\begin{aligned} \mathcal{R}_{0(0)} &= \max_i\left\{ \sqrt{\beta^h \frac{b}{\mu_i} \beta^v \frac{b}{r} \frac{p^v}{p^v + \mu_i} \frac{1}{N^h} \frac{\Lambda_i}{\mu_i}} \right\} \\ &= \begin{cases} \mathcal{R}_{00}, & \text{Uncontrolled system} \\ \mathcal{R}_{0N}, & \text{Less than } 100\% \text{ compliant system} \\ \mathcal{R}_{0C}, & 100\% \text{ compliant system.} \end{cases} \end{aligned} \tag{14}$$

The eigenvectors $\mathbf{I}^h_{(0)}$ and $\mathbf{I}^v_{(0)}$ are not uniquely determined by Eqs (7) and (8) due to degeneracy in the eigenspaces of the second generation matrices when $\omega = 0$ (see S1 Appendices Sec. 4.4). Specifically, any vector which is distributed entirely within any subset of the non-compliant sites will be an eigenvector with eigenvalue $\mathcal{R}^2_{0(0)}$. Consequently, we know that the worst

case distributions of infectious vectors and hosts will lie entirely within the non-compliant sites in the no-hopping limit, but we have no way of distinguishing which, if any, of all possible distributions are more "important" or "correct" practically, in terms of biological meaning and control strategies. This ambiguity can be resolved to some extent by considering perturbations to the no-hopping case caused by small but non-zero hopping rates.

**Finitely slow hopping.** For the case of slow but non-zero hopping, we analyze our system using degenerate perturbation theory. Here, we give the only the necessary definitions and mathematical formulae from which we obtain results. Details of the derivation are provided in S1 Appendices Sec. 5, and details of the perturbation formalism in general can be found at a utilitarian level in Ref. [48], or at a more rigorous mathematical level in Ref. [49]. Perturbation theory will provide accurate results for our system when the parameters $\omega/\mu_0$ and $\omega/(\mu_0 + p^\nu)$ are much smaller than unity. Under this assumption, one can Taylor expand the terms in Eqs (7) and (8) to first order in $\omega$ about $\omega = 0$:

$$(\mathcal{R}_{0(0)} + \delta\mathcal{R}_0)^2(\mathbf{I}^h_{(0)} + \delta\mathbf{I}^h) \quad = \quad (\mathcal{M}_{(0)} + \delta\mathcal{M})(\mathbf{I}^h_{(0)} + \delta\mathbf{I}^h), \tag{15}$$

$$(\mathcal{R}_{0(0)} + \delta\mathcal{R}_0)^2(\mathbf{I}^\nu_{(0)} + \delta\mathbf{I}^\nu) \quad = \quad (\underline{\mathcal{M}}_{(0)} + \delta\underline{\mathcal{M}})(\mathbf{I}^\nu_{(0)} + \delta\mathbf{I}^\nu), \tag{16}$$

Here, the subscript (0) refers to quantities evaluated in the no-hopping limit $\omega = 0$, and the perturbations signified by $\delta$ are linear in $\omega$, where all terms quadratic and higher in $\omega$ are discarded. $\mathcal{R}_{(0)}$ is given in Eq (14), and $\mathbf{I}^\nu_{(0)}$ and $\mathbf{I}^h_{(0)}$ are non-zero in only the non-compliant sites, but are otherwise yet to be determined.

To solve the perturbed eigenvalue problem, assume that exactly $J \geq 1$ sites are non-compliant, and let $\{i_1, i_2, \ldots, i_J\}$ denote their indices. In S1 Appendices Sec. 5, we show that the solutions to the perturbed eigenvalue problem are ultimately determined by the spectrum of the $J \times J$-dimensional matrix $\mathcal{W}$ defined by the following elements:

$$\mathcal{W}_{jk} \quad = \quad \begin{cases} -\kappa \, \mathrm{degN}(i_j) - \kappa\,\xi\,\mathrm{degC}(i_j), & j = k \\ \kappa, & \text{If sites } i_j \text{ and } i_k \text{ are nearest neighbors} \\ 0, & \text{otherwise,} \end{cases}$$

Here, $\mathrm{degC}(n)$ and $\mathrm{degN}(n)$ denote the numbers of compliant and non-compliant nearest neighbors connected to a site $n$, respectively, and the dimensionless constants $\kappa$ and $\xi$ are given by

$$\kappa \quad = \quad \frac{1}{2}\mathcal{R}_{0N}\left[\left(\frac{\mu_N}{\mu_0}\right)^{-1} + \left(\frac{p^\nu}{\mu_0} + \frac{\mu_N}{\mu_0}\right)^{-1}\right]\frac{\omega}{\mu_0}, \tag{17}$$

and

$$\xi \quad = \quad 1 + \frac{1}{1 + \frac{\mu_N}{p^\nu + \mu_N}}\left(1 - \frac{\Lambda_C/\mu_C}{\Lambda_N/\mu_N}\right). \tag{18}$$

The solutions to the perturbed eigenvalue problem are summarized as follows. The perturbation $\delta\mathcal{R}_0$ is the largest eigenvalue of $\mathcal{W}$, is non-positive, and depends only on $\xi$, the spatial configuration of non-compliant sites, and linearly on $\kappa$. The $J$-dimensional eigenvectors $\boldsymbol{\alpha}$ defined by the relation

$$\delta\mathcal{R}_0\boldsymbol{\alpha} \quad = \quad \mathcal{W}\boldsymbol{\alpha}, \tag{19}$$

can be taken to be non-negative and normalized. The components of $\boldsymbol{\alpha}$ determine $\mathbf{I}_{(0)}^{h}, \mathbf{I}_{(0)}^{v}, \delta\mathbf{I}^{h}$, and $\delta\mathbf{I}^{v}$ as follows:

$$
\begin{aligned}
\mathbf{I}_{(0)}^{h} &= \mathbf{I}_{(0)}^{v} \\
&= \sum_{k=1}^{J} \alpha_k \mathbf{e}_{i_k},
\end{aligned}
\tag{20}
$$

$$
\delta\mathbf{I}^{h} = -\sum_{k=1}^{J} \sum_{\substack{j=1 \\ j\neq i_1,i_2,\ldots,i_J}}^{\mathcal{N}^2} \frac{\delta M_{j\,i_k}}{\mathcal{R}_{0N}^2 - \mathcal{R}_{0C}^2} \alpha_k \mathbf{e}_{i_k} + \sum_{\substack{j=1 \\ j\neq i_1,i_2,\ldots,i_J}}^{\mathcal{N}^2} \sum_{k=1}^{J} \frac{\delta M_{j\,i_k}}{\mathcal{R}_{0N}^2 - \mathcal{R}_{0C}^2} \alpha_k \mathbf{e}_j,
\tag{21}
$$

$$
\delta\mathbf{I}^{v} = -\sum_{k=1}^{J} \sum_{\substack{j=1 \\ j\neq i_1,i_2,\ldots,i_J}}^{\mathcal{N}^2} \frac{\delta\underline{M}_{j\,i_k}}{\mathcal{R}_{0N}^2 - \mathcal{R}_{0C}^2} \alpha_k \mathbf{e}_{i_k} + \sum_{\substack{j=1 \\ j\neq i_1,i_2,\ldots,i_J}}^{\mathcal{N}^2} \sum_{k=1}^{J} \frac{\delta\underline{M}_{j\,i_k}}{\mathcal{R}_{0N}^2 - \mathcal{R}_{0C}^2} \alpha_k \mathbf{e}_j.
\tag{22}
$$

Here, $\mathbf{e}_k$ denotes the standard unit vector in $\mathbb{R}^{\mathcal{N}^2}$ which points along the $k^{th}$ dimension. The first double summations in $\delta\mathbf{I}^{h}$ and $\delta\mathbf{I}^{v}$ are the perturbations to $\mathbf{I}_{(0)}^{h}$ and $\mathbf{I}_{(0)}^{v}$ within the sites occupied by $\mathbf{I}_{(0)}^{h}$ and $\mathbf{I}_{(0)}^{v}$ (all of which are non-compliant), while the second double summations give the perturbations at the sites not occupied by $\mathbf{I}_{(0)}^{h}$ and $\mathbf{I}_{(0)}^{v}$. The matrix element perturbations $\delta\mathcal{M}_{ij}$ and $\delta\underline{\mathcal{M}}_{ij}$ are non-zero only if $i = j$ or if site $i$ and site $j$ are connected (see S1 Appendices Sec. 5), and so the eigenvector perturbations $\delta\mathbf{I}^{h}$ and $\delta\mathbf{I}^{v}$ effectively transport the unperturbed eigenvectors from connected blocks of non-compliant sites out into the corresponding bordering compliant sites.

The above mathematics can be interpreted by recognizing that $\mathcal{W}$ is the matrix for a linear ordinary differential equation describing the population dynamics of a nearest neighbor random walk through the non-compliant sites at (unit-less) hopping rate $\kappa$, with reflecting boundary conditions imposed at the boundaries between connected blocks of non-compliant sites and the surrounding compliant sites, together with site-dependent (unit-less) death rates proportional to $\kappa\xi$. Specifically,

$$
\mathcal{W} = -\kappa\mathcal{L}^* - \kappa\,\xi\,\mathcal{D},
\tag{23}
$$

where $\mathcal{L}^*$ is a nearest neighbor hopping Laplacian matrix and $\mathcal{D}$ is a diagonal death rate matrix with components $\mathcal{D}_{jj} = \mathrm{degC}(i_j)$. Thus, by relabeling the indices of the non-compliant sites, $\mathcal{W}$ can always be brought into block diagonal form, where each matrix block corresponds to a distinct block of connected non-compliant sites. We denote the block-diagonalized matrix by $\widetilde{\mathcal{W}}$ for reference. Each matrix block of $\widetilde{\mathcal{W}}$ will have a largest real non-positive eigenvalue which represents a candidate value for $\delta\mathcal{R}_0$, and the actual value for $\delta\mathcal{R}_0$ is found by calculating all candidate values for all matrix blocks and taking the largest (e.g. smallest-in-magnitude) result.

For a hypothetical population distribution evolving in time under $\mathcal{W}$, within each block of connected non-compliant sites, the action of $-\kappa\mathcal{L}^*$ conserves total population levels, while the action of $-\kappa\,\xi\mathcal{D}$ decreases the total population level via non-zero death rates within the blocks' border sites. For each matrix block of $\widetilde{\mathcal{W}}$, the magnitude of its largest eigenvalue (e.g. the magnitude of that matrix block's candidate $\delta\mathcal{R}_0$ value) gives the characteristic asymptotic rate of death for a hypothetical population distributed in the corresponding non-compliant block, thus implying that the value of $\delta\mathcal{R}_0$ is determined by the non-compliant block (or blocks) with

the smallest characteristic rate of death under the dynamics driven by $\mathcal{W}$. Within a non-compliant block, the death rate under $\mathcal{W}$ at any one site increases with the number of nearest neighbor connections from that site to the compliant sites that border the block, so one intuitively expects to associate smaller characteristic rates of death to the non-compliant blocks that have a reduced capacity to leak distributions out through their boundaries into the surrounding compliant sites (that is, blocks that are more tightly clustered together, with smaller boundaries and greater ratios of inner non-compliant connections to outer compliant connections). The mathematics of perturbation theory thus indicate that, for a given set of compliant and non-compliant control efficacies and a given neighborhood compliance structure, the reduction in $\mathcal{R}_0$ induced by small non-zero mosquito hopping (e.g. the value of $\delta\mathcal{R}_0$) will be determined by the larger and more tightly clustered non-compliant blocks. Correspondingly, Eqs (19) and (20) imply that $\mathbf{I}_{(0)}^v$ and $\mathbf{I}_{(0)}^h$ will lie entirely in the larger more tightly clustered non-compliant blocks.

## Optimized control

In this section, we formulate the mathematics needed to determine cost-optimal door-to-door and aerial spray application strategies for reducing the model $\mathcal{R}_0$ to unity. Our goal is to translate the mathematical formalism developed for our $\mathcal{R}_0$ analysis into actionable control advice for the real world. Specifically, we wish to determine the degree to which vector hopping rates, compliance levels, and compliance clustering influence the efficacy and optimal cost-effectiveness of door-to-door control. We are particularly interested in determining conditions under which door-to-door control becomes more or less cost-effective than area-wide aerial spraying. Here, we assume a $10 \times 10$ neighborhood consisting of 100 houses. We note that although our focus is on preventative control, the cost-optimal $\mathcal{R}_0$ reduction strategies derived in this section are also effective strategies for containing the total size of an active outbreak. It is important to note, however, that although effective, cost-optimal strategies for the control goal $\mathcal{R}_0 = 1$ are unlikely to also be cost-optimal for the control goal of total outbreak reduction. This is due in particular to the fact that the total outbreak size depends on the full disease dynamics and initial conditions, while $\mathcal{R}_0$ depends only on the dynamics linearized about the disease-free equilibrium. Details and examples are provided in S1 Appendices Sec. 8.

**Control modeling methodology.** Controls representing vector management strategies are incorporated into our model as time-independent, site-dependent increases and decreases in vector death and emergence rates, respectively, as defined in Eqs (5) and (6). Door-to-door adulticide and larval source reduction uniformly increase death rates and reduce emergence rates, respectively, in compliant sites only, while area-wide adulticide spray uniformly increases death rates in all sites. Defining the effects of control in this manner is useful for associating changes in model parameters, and therefore predictions for outbreak potential via $\mathcal{R}_0$, with notions of control effort. However, controls modeled as fixed variations in model parameters can only serve as approximate representations of the effects of real-world vector-management strategies, which are, in reality, applied in discontinuous impulses and have finite efficacy times. In order to relate reductions in $\mathcal{R}_0$ to real-world control actions and costs, one must specify a scheme by which fixed variations in death and emergence rates approximate real-world discontinuous control impulses.

In this paper, we apply the control methodology established in Ref. [28]; the modified death and emergence rates in our model describe the time-averaged effects of real-word vector control strategies applied repeatedly at fixed frequencies, where potential resonance-like synergistic effects arising from the interaction and relative timing of controls are ignored. Results obtained from this control modeling methodology are intended to give public health

workers a basic intuitive sense of approximately how often different classes of controls need to be applied in order to suppress vector-borne disease outbreaks in a neighborhood setting. This type of utilitarian information can serve as one of the many scientific and practical considerations which together inform public policy decisions for vector management. This control modeling methodology is not intended for use in recommending specifically timed impulse control protocols tuned to specific biological details, as results obtained at such detailed levels are likely to depend sensitively on model structure and assumptions which can not easily or reliably capture biological complexity and real-world constraints encountered in the field.

In S1 Appendices Sec. 6, we relate the emergence and death rates in compliant and non-compliant sites to the aerial spray and door-to-door application frequencies, denoted $f_A$ and $f_D$, respectively, in terms of real-world control parameters like percent knock down and efficacy decay time. These parameters are given real-world values based on experimental observations. The aerial spray application frequency represents a fixed rate at which airplanes are sent out to apply aerial adulticide spray, and the door-to-door application frequency represents a fixed rate at which teams of workers are sent out to apply residual barrier spray and reduce larval sources in yards. When a control is applied regularly at a fixed rate, it is reasonable to expect the associated cost of control to be roughly proportional to the application frequency. Door-to-door control requires small teams of government or health agency employees to visit every site in the neighborhood so that they may request property access from the sites' residents, and teams will spend additional time at compliant sites implementing residual barrier spray and larval source reduction, so greater numbers of man-hours are required to deliver door-to-door control to neighborhoods with higher levels of compliance. We therefore assume the following daily cost of control $C$ which depends linearly on the application frequencies $f_A$ and $f_D$:

$$C \quad = \quad c_A f_A + (c_{DC} f + c_{DN}(1 - f))f_D. \tag{24}$$

In the above equation, $c_A$ is the cost per application of adulticide aerial spray applied to the entire neighborhood, $c_{DC}$ is the cost for applying door-to-door control to a 100-house neighborhood at 100% compliance, $c_{DN}$ is the labor cost for applying door-to-door control to a 100-house neighborhood at 0% compliance, and $f$ is the fraction of compliant sites in the neighborhood. The numerical values of $c_A$, $c_{DC}$, and $c_{DN}$ used in our analysis are based on expert opinion (Kevin Caillouët, personal communication), and are given in S1 Appendices Table A.

**Control problem formulation.** We consider the following control problem: for combined aerial spray and door-to-door control strategies, aerial spray only, and door-to-door control only, find the application frequencies $f_A$ and $f_D$ which suppress outbreak potential by reducing $\mathcal{R}_0$ to unity while minimizing the cost of control $C$, subject to control bounds $f_A \in [0 \text{ day}^{-1}, 1 \text{ day}^{-1}]$ and $f_D \in [(20 \text{ year})^{-1}, 1 \text{ day}^{-1}]$. The upper bounds on $f_A$ and $f_D$ limit control applications to occur at a rate of at most once per day. Even if unlimited resources were available, societal and logistical concerns would likely prohibit government or health agency employees from applying pesticides and invading yards more than once per day. The lower bound on $f_D$ is close to enough zero for all practical purposes, but is set to a non-zero number so that certain integrals can be computed over a finite range (see door-to-door adulticide in S1 Appendices Sec. 6). The cost function in Eq (24) is linear in the application frequencies $f_A$ and $f_D$, so our control problem is a bounded linear optimization problem subject to a non-linear constraint $\mathcal{R}_0 = 1$. We solve all optimization problems numerically using the *fmincon* function in *Matlab*

*R2017a*. Note that in using a local optimization function like *fmincon*, one is required to make an initial guess for the optimal application frequencies, and a poorly chosen initial guess may produce sub-optimal solutions.

For our numerical analysis, we use the model parameter values in Table 1 and control parameter values given in S1 Appendices Table A. Under these parameters, for any hopping rate or compliance distribution, it is always possible to find application frequencies $f_A$ and $f_D$ within the control bounds which reduce $\mathcal{R}_0$ to unity for aerial spray only control and for combined aerial spray and door-to-door control strategies. For door-to-door only control, however, there exist hopping rate / compliance distribution combinations for which $\mathcal{R}_0$ cannot be brought to one for any $f_D$ less than or equal to 1 day$^{-1}$. In such circumstances, we refer to the system as being uncontrollable under door-to-door control alone. For an uncontrollable system, as long as there is at least one compliant site in the neighborhood, door-to-door only control will provide some benefit by reducing $\mathcal{R}_0$ to an extent, even if not all the way to one. In this case, the optimal door-to-door only control solution will be $f_D$ = 1 day$^{-1}$, meaning that door-to-door only control should be applied as often as possible in order to bring $\mathcal{R}_0$ as close as possible to one. If there are no compliant sites in the neighborhood, door-to-door only control will have no influence over $\mathcal{R}_0$, and the optimal door-to-door only control solution will be to never spend money on door-to-door control, so $f_D$ = 0 day$^{-1}$.

To obtain numerical results, we first solve the optimized control problem for the special cases of no-hopping, infinitely fast hopping, and homogeneous systems by using the explicit analytic expressions obtained for $\mathcal{R}_0$ in Eqs (10), (12), and (14). For these special cases, compliance spatial structure and boundary conditions are irrelevant. Second, we solve the optimized control problem for four specific compliance distributions (20% compliance and 60% compliance, both randomly dispersed and highly clustered) under a range of hopping rates by numerically obtaining $\mathcal{R}_0$ as the largest eigenvalue of either second generation matrix. Here, boundary conditions must be specified, so we employ both periodic and reflecting boundary conditions in order to compare the effects of boundary choice on model behavior. Third, we determine the average optimized costs and application frequencies for door-to-door control alone as a function of percent compliance by solving the optimization problem over large numbers of random compliance distributions at compliance level between 0% and 100%, assuming periodic boundary conditions for simplicity. To randomly generate compliance distributions, we apply the landscape pattern with random clusters algorithm of Saura and Martínez-Milláan [50]. This algorithm produces random compliance distributions at any desired percent compliance, such that the degree of compliance clustering is specified by a clustering parameter $P \in [0, 1]$. When $P = 0$, the compliant sites are randomly dispersed around the neighborhood with no spatial correlation, and as $P$ approaches 1, the compliant sites tend gather into a single random cluster. When $P = 0.59$, about half of the compliant sites tend to clump into a single random cluster, while the other half tend to gather in clusters of smaller size [50]. For these random configuration simulations, we select the hopping rates $0.5\mu_0$, $\mu_0$, and $5\mu_0$, and for each hopping rate we generate 200 random dispersed distributions at $P = 0$ and 200 random clustered distributions at $P = 0.59$ for all levels of compliance between 0% and 100%. For each compliance level / hopping rate / clustering setting, we find the optimized door-to-door only control costs and application frequencies for all generated configurations, and then calculate the average optimized cost of control and application frequency. In addition, we calculate the fraction of configurations controllable under door-to-door control alone, as well as the fraction of configurations that are both controllable and cheaper to control with door-to-door control than with aerial spray.

## Results

### The basic reproduction number

To aid in interpreting our results, Table 2 summarizes the meanings of relevant parameters and quantities used in our figures.

**No-hopping, infinitely fast hopping, and homogeneous systems.** In Fig 2, we plot the analytic expressions for $\mathcal{R}_0$ derived for infinitely slow hopping, infinitely fast hopping, and homogeneous systems, assuming model parameters given in Table 1. Fig 2a represents the single-site, homogeneous system, and no-hopping $\mathcal{R}_0$ values as a function of relative adulticide strength $\mu_i/\mu_0$ and relative larval source reduction strength $\Lambda_i/\Lambda_0$ (we plot in terms of relative strengths so that the results are presented most transparently in terms of control effort). For the case of a single site, $\mu_i$ and $\Lambda_i$ represent the site's controlled death and emergence rates, and for the case of a homogeneous system, $\mu_i$ and $\Lambda_i$ represent the controlled death and emergence rates which are uniform over the entire system. For an inhomogeneous system, Fig 2a represents the no-hopping basic reproduction number when $\mu_i$ and $\Lambda_i$ represent non-compliant death and emergence rates. We assume that non-compliant sites are only subject to an adulticide spray which does not decrease vector emergence rates below their natural value $\Lambda_0$, so the relevant values of $\mathcal{R}_0$ are given along the top of Fig 2a where $\Lambda_i/\Lambda_0 = 1$. In general we see that $\mathcal{R}_0$ decreases as $\mu_i/\mu_0$ increases from zero to infinity, and that $\mathcal{R}_0$ increases as $\Lambda_i/\Lambda_0$ increases from zero to one, thus implying that $\mathcal{R}_0$ generally decreases with increasing control efficacy. The red line in Fig 2a represents $\mathcal{R}_0 = 1$ so parameter values to the right of and below this line will effectively suppress outbreak potential for our model parameters in a single-site system, homogeneous system, or no-hopping system. The case of no control is given in the upper left-hand corner of Fig 2a, where $\mathcal{R}_0 = 1.89$. Note that for the case of negligible intrinsic incubation period $p^v \to \infty$, our parameters give $\mathcal{R}_0 = 2.67$ under uncontrolled conditions.

**Table 2. Quantities used in the analyses of the basic reproduction number results.**

| Quantity | Meaning | Notes |
|---|---|---|
| $\Lambda_i/\Lambda_0$ | Relative measure of larval source reduction control effort in an isolated patch | Weak control when close to 1, decreases with increasing control strength |
| $\mu_i/\mu_0$ | Relative measure of adulticide control effort in an isolated patch | Weak control when close to 1, increases with increasing control strength |
| $\mu_N/\mu_0$ | Relative measure of adulticide aerial spray control efficacy in a non-compliant patch | Weak control when close to 1, increases with increasing control strength |
| $\mu_N/p^v$ | Alternative relative measure of adulticide aerial spray control efficacy in a non-compliant patch | Increases with increasing control strength |
| $\frac{\Lambda_C/\mu_C}{\Lambda_N/\mu_N}$ | Relative compliant patch combined adult and larval control efficacy to non-compliant patch control efficacy | Equal compliant and non-compliant control efficacy when equal to 1, decreases as compliant control efficacy increases |
| $\omega/\mu_0$ | Average number of hops taken by a vector over the average natural lifespan | Fast hopping regime when much greater than 1, slow hopping regime when much less than 1 |
| $\mathcal{R}_{0\infty}$ | Infinite hopping basic reproduction number | None |
| $\mathcal{R}_{0(0)}$ | No-hopping basic reproduction number | None |
| $\delta\mathcal{R}_0$ | Perturbation to the no-hopping basic reproduction number induced by small hopping rates | Function of only the size and spatial structure of non-compliant blocks, the parameter $\xi$, and linear dependence on the parameter $\kappa$ |
| $\xi$ | Abstract unit-less "death rate" in non-compliant blocks which determines $\delta\mathcal{R}_0$ | Function of only $\mu_N/\mu_0$ and $\frac{\Lambda_C/\mu_C}{\Lambda_N/\mu_N}$ |
| $\kappa$ | Abstract unit-less "hopping rate" in non-compliant blocks which determines $\delta\mathcal{R}_0$ | Function of uncontrolled model parameters, non-compliant adulticide control efficacy $\mu_N/\mu_0$, and linear dependence on $\omega/\mu_0$ |

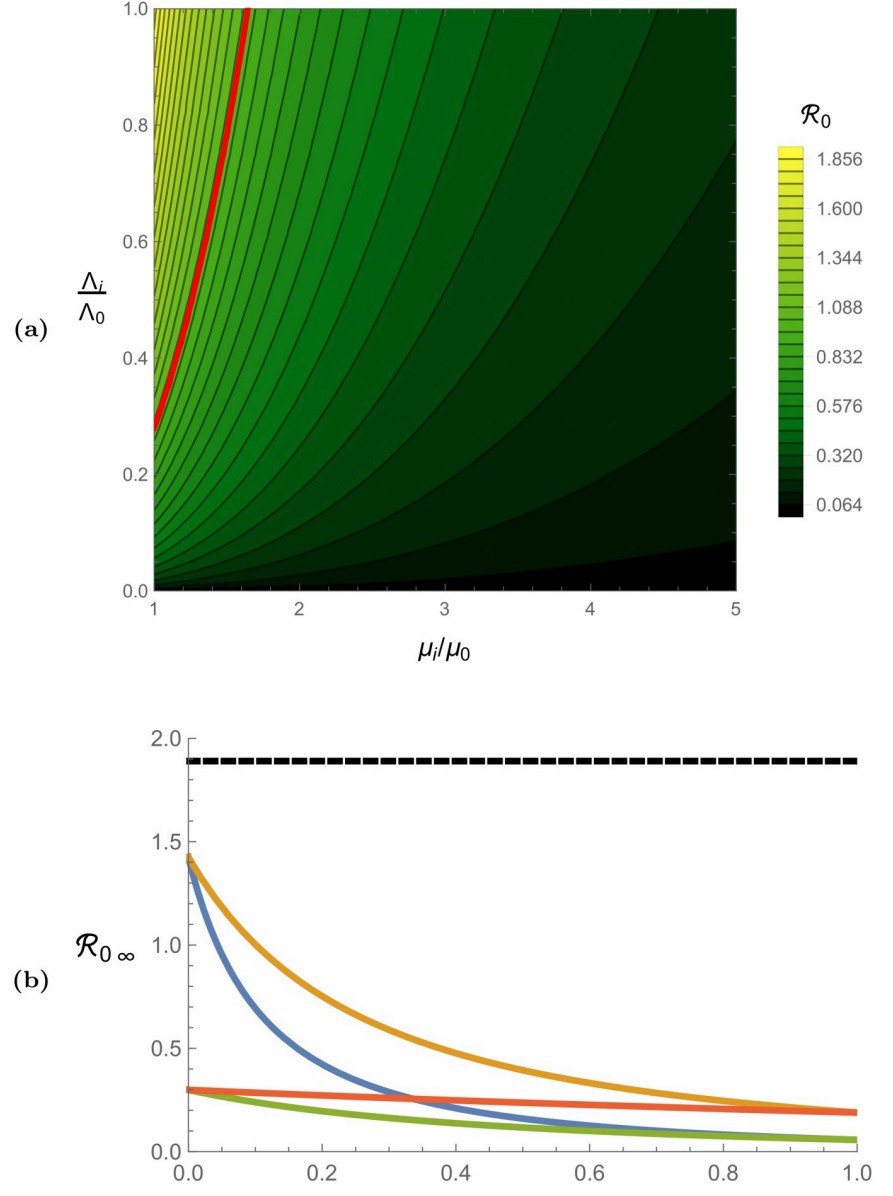

**Fig 2.** (a) Single-site basic reproduction number values as a function of the relative controlled death rate $\mu_i/\mu_0$ and relative controlled emergence rate $\Lambda_i/\Lambda_0$, assuming model parameters given in Table 1. These values also represent the homogeneous system basic reproduction numbers when $\mu_i$ and $\Lambda_i$ are uniform across the neighborhood, as well as the no-hopping basic reproduction numbers when $\mu_i$ and $\Lambda_i$ are the non-compliant death and emergence rates. Control efficacy increases along the $\mu_i/\mu_0$ axis and decreases along the $\Lambda_i/\Lambda_0$ axis. The case of no control is represented by the upper left corner where $\mathcal{R}_0 = 1.89$, and the red line indicates $\mathcal{R}_0 = 1$. Control efficacy for non-compliant sites is represented along the top of the plot where $\Lambda_i = \Lambda_0$. (b) Infinitely fast hopping basic reproduction number values as a function of percent compliance, assuming model parameters given in Table 1. The **black** line represents the uncontrolled value $\mathcal{R}_{0\infty} = 1.89$, and the different color curves represent varying degrees of control efficacy in compliant and non-compliant sites. Blue represents strong compliant efficacy and weak non-compliant efficacy ($\mu_N = 1.25\mu_0$, $\mu_C = 10\mu_0$, $\Lambda_C = .5\Lambda_0$), orange represents moderate compliant efficacy and weak non-compliant efficacy ($\mu_N = 1.25\mu_0$, $\mu_C = 5\mu_0$, $\Lambda_C = .75\Lambda_0$), green represents strong compliant efficacy and moderate non-compliant efficacy ($\mu_N = 4\mu_0$, $\mu_C = 10\mu_0$, $\Lambda_C = .5\Lambda_0$), and red represents moderate compliant efficacy and moderate non-compliant efficacy ($\mu_N = 4\mu_0$, $\mu_C = 5\mu_0$, $\Lambda_C = .75\Lambda_0$).

Fig 2b shows the infinite hopping basic reproduction number $\mathcal{R}_{0\infty}$ under our model parameters as a function of percent compliance, assuming various compliant and non-compliant control efficacies. At zero percent compliance, door-to-door control has no effect on the system, so the corresponding $\mathcal{R}_{0\infty}$ values in Fig 2b represent aerial control only and are equivalent to the homogeneous system $\mathcal{R}_0$ values. In general, $\mathcal{R}_{0\infty}$ decreases as a function of percent compliance, and it decreases more rapidly when the relative efficacy of compliant control to non-compliant control is larger.

**Finitely slow hopping.**   Figs 3, 4, and 5 together give the perturbation analysis numerical results for small non-zero hopping rates. For small hopping rates (meaning $\omega/\mu_0 \ll 1$), the basic reproduction number is given by $\mathcal{R}_0 = \mathcal{R}_{0(0)} + \delta\mathcal{R}_0$, where $\mathcal{R}_{0(0)}$ is the no-hopping basic reproduction number (the values of which are given in Fig 2a), and $\delta\mathcal{R}_0$ is the perturbation induced by small hopping. The value of $\delta\mathcal{R}_0$ depends only on the number and spatial structure of the non-compliant sites, the parameter $\kappa$, and the parameter $\xi$. The perturbation $\delta\mathcal{R}_0$ is proportional to $\kappa$, and $\kappa$ is a function of the uncontrolled model parameters, the relative strength of non-compliant control $\mu_N/\mu_0$, and is proportional to $\omega/\mu_0$. In Fig 3a, we plot $\kappa/(\omega\mu_0^{-1})$ as a function of $\mu_N/\mu_0$ for the model parameters given in Table 1. We scale $\kappa$ by $\omega/\mu_0$ to remove the linear dependence on this parameter so that the results presented apply as universally as possible; the plot is applicable for any hopping rate, while the plotted quantity is unitless and therefore independent of the units used for time. We plot $\kappa$ as a function of $\mu_N/\mu_0$ so that the results are presented most transparently in terms of control effort. This figure shows that $\kappa$ decreases to zero as non-compliant control efficacy increases, and reaches its maximum value 2.23 for the case of no non-compliant control $\mu_N = \mu_0$ (meaning no aerial spray). We note that in the limit of negligible extrinsic incubation period $p^\nu \to \infty$, our model parameters give a $\kappa/(\omega\mu_0^{-1})$ curve nearly identical to the one in Fig 3a.

The parameter $\xi$ itself depends many model parameters, but can be written as a function of only two dimensionless quantities related to control effort: the relative efficacy of compliant control to non-compliant control $\frac{\Lambda_C/\mu_C}{\Lambda_N/\mu_N}$ (e.g. the ratio of the equilibrium vector population in an isolated compliant site to that of an isolated non-compliant site), and the non-compliant adulticide efficacy measured relative to the extrinsic incubation period $\mu_N/p^\nu$ (e.g. a measure of adulticide control effort). In Fig 3b, we plot $\xi$ as a function of $\frac{\Lambda_C/\mu_C}{\Lambda_N/\mu_N}$ and $\mu_N/p^\nu$. For the model parameters considered in Table 1, we have $p^\nu = \mu_0$, and all possible values of $\xi$ will fall on or to the right of the solid black line in Fig 3b. The solid black line corresponds to the case of no non-compliant control (no aerial spray) when $\mu_0 = p^\nu$, and $\xi$ decreases from 5/3 to 1 along this line as $\frac{\Lambda_C/\mu_C}{\Lambda_N/\mu_N}$ increases. In the limit of infinitely strong non-compliant control $\mu_N/p^\nu \to \infty$, $\xi$ decreases from 3/2 to 1 as $\frac{\Lambda_C/\mu_C}{\Lambda_N/\mu_N}$ increases. In the limit of negligible extrinsic incubation period $p^\nu \to \infty$, $\xi$ decreases from 2 to 1 as $\frac{\Lambda_C/\mu_C}{\Lambda_N/\mu_N}$ increases. Generally, $\xi$ will fall somewhere in the interval [1, 2], and will be larger when non-compliant control is weak in comparison to compliant control and in comparison to $p^\nu$.

In Fig 4, we draw a number of clustering arrangements for non-compliant blocks of sizes one through six sites, and in Fig 5, we plot the corresponding values of $\delta\mathcal{R}_0/\kappa$ induced by those non-compliant blocks as a function of $\xi$, assuming that the blocks pictured are completely surrounded by compliant sites (e.g. are not a part of the neighborhood boundary where the choice of periodic or reflecting boundary conditions will matter). We scale $\delta\mathcal{R}_0$ by $\kappa$ in Fig 5 to remove the linear dependence on this parameter so that the results apply universally for any value of $\kappa$ (larger $\kappa$ values will uniformly increase the rates of decline of the curves in Fig 5). If more than one non-compliant configuration in Fig 4 is present in a neighborhood, the corresponding values of $\delta\mathcal{R}_0$ indicated by Fig 5 will represent candidate $\delta\mathcal{R}_0$ values, and

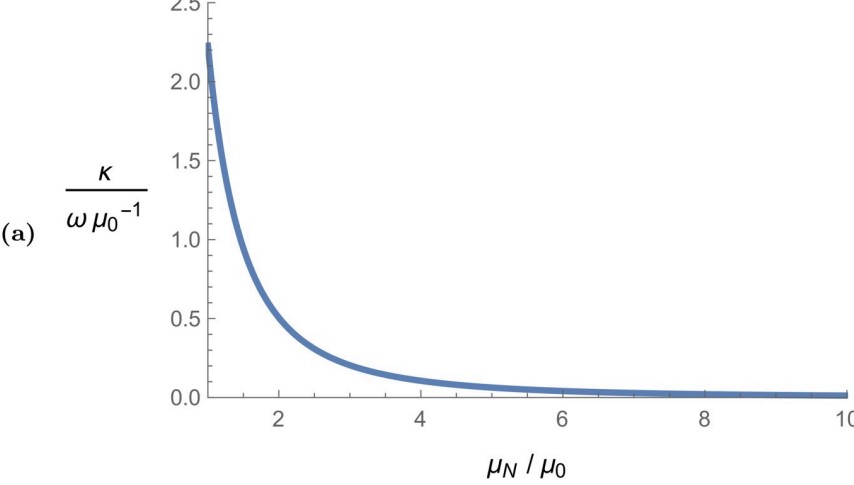

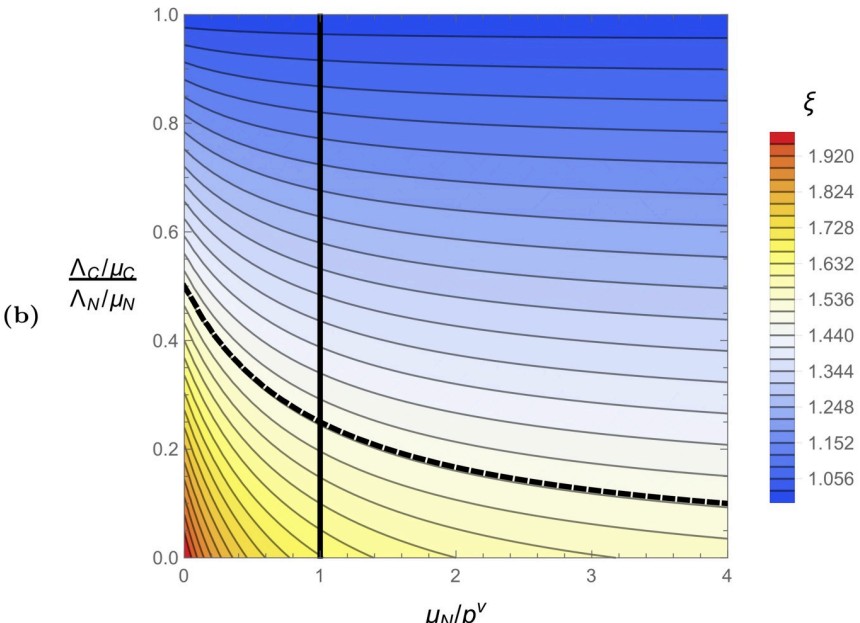

**Fig 3.** (a) Linear scaling factor of $\kappa$ with respect to $\frac{\omega}{\mu_0}$ as a function of the relative non-compliant death rate $\frac{\mu_N}{\mu_0}$. Weakly efficacious non-compliant control (e.g. weakly efficacious aerial spray) is represented in the regime $\frac{\mu_N}{\mu_0} \approx 1$, and strongly efficacious non-compliant control is represented by large values of $\frac{\mu_N}{\mu_0}$. (b) Values of $\xi$ as a function of the relative efficacy of compliant control to non-compliant control $\frac{\Lambda_C/\mu_C}{\Lambda_N/\mu_N}$ and non-compliant control efficacy $\mu_N/p^\nu$. Relative compliant efficacy decreases along the vertical axis, and non-compliant efficacy increases along the horizontal axis. The limit of negligible extrinsic incubation period $p^\nu \to \infty$ is represented by the vertical axis $\mu_N/p^\nu = 0$. For the model parameters considered in Table 1, we have $\mu_0 = p^\nu$, so the smallest possible value of $\mu_N/p^\nu$ is 1 (which represents no control in non-compliant sites, e.g. no aerial spray), and all possible values of $\xi$ will fall to the right of the solid black $\mu_N/p^\nu = 1$ line. Along this line, $\xi$ decreases from 5/3 to 1 as $\frac{\Lambda_C/\mu_C}{\Lambda_N/\mu_N}$ increases. In the limit of infinitely strong non-compliant control $\mu_N/p^\nu \to \infty$, the dotted black line representing $\xi = 1.5$ asymptotes to the $\mu_N/p^\nu$ axis.

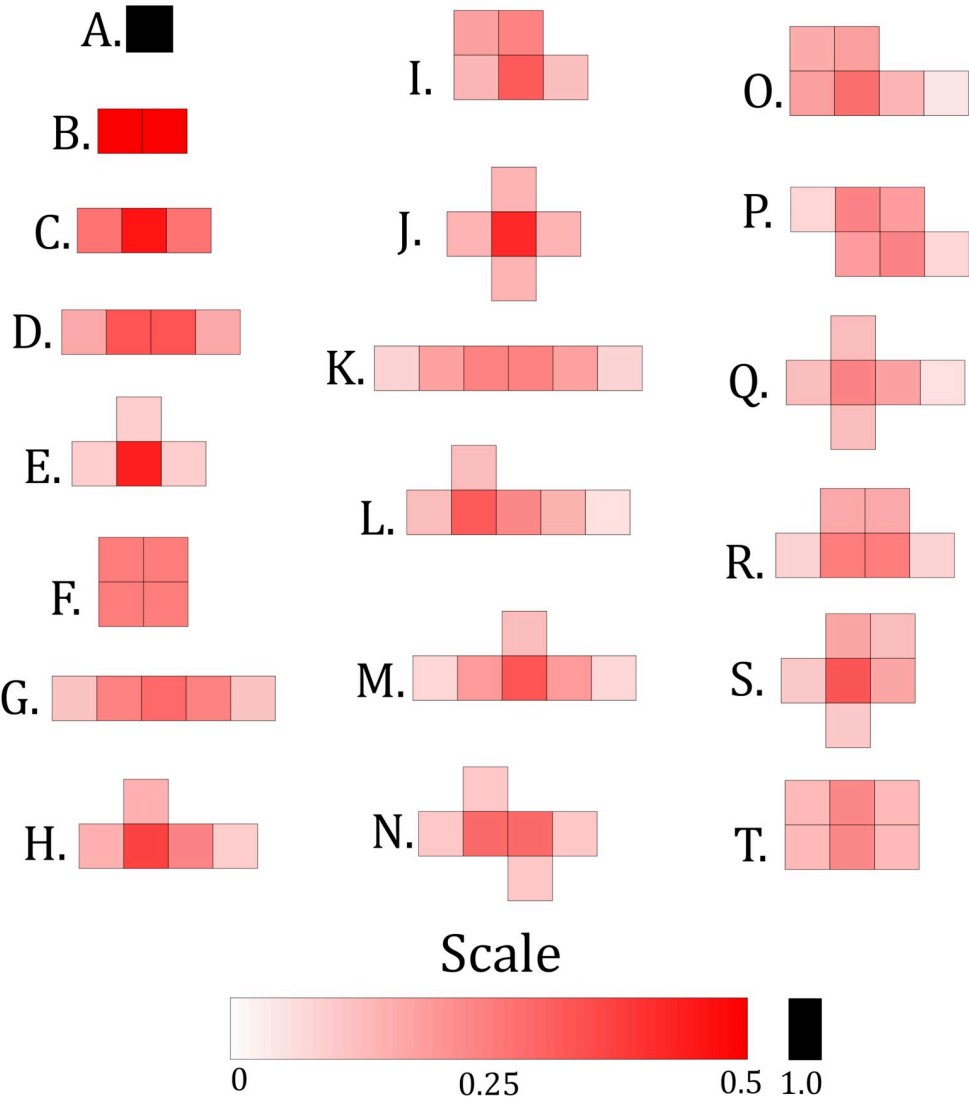

**Fig 4. Various possible arrangements of varying sizes of non-compliant blocks.** The shading within a particular configuration indicates the distributions of the unperturbed eigenvectors $\mathbf{I}_{(0)}^h$ and $\mathbf{I}_{(0)}^v$ as determined by first order perturbation analysis when that configuration is the unique configuration which determines $\delta\mathcal{R}_0$ in a neighborhood, assuming $\xi = 1.5$ and that the pictured blocks are not part of the neighborhood boundary. Any configuration of size one through six can be obtained from one of the pictured configurations through a number of bending symmetry operations.

the actual value of $\delta\mathcal{R}_0$ will be determined by the block with largest candidate value (meaning the candidate value which is smallest in magnitude due to all candidate values being non-positive). The shading within any one particular configuration in Fig 4 shows the distributions of the unperturbed eigenvectors $\mathbf{I}_{(0)}^h$ and $\mathbf{I}_{(0)}^v$ as determined by first order perturbation analysis when that configuration gives the unique largest candidate $\delta\mathcal{R}_0$ value in a neighborhood, assuming $\xi = 1.5$ and that the pictured blocks are not part of the neighborhood boundary. For a given configuration, we see that $\mathbf{I}_{(0)}^h$ and $\mathbf{I}_{(0)}^v$ tend to peak and decay away from the sites with the fewest number of connections to the configuration border. We note that for values of $\xi$ other than 1.5, the distributions of $\mathbf{I}_{(0)}^h$ and $\mathbf{I}_{(0)}^v$ are visually similar to the the distributions in

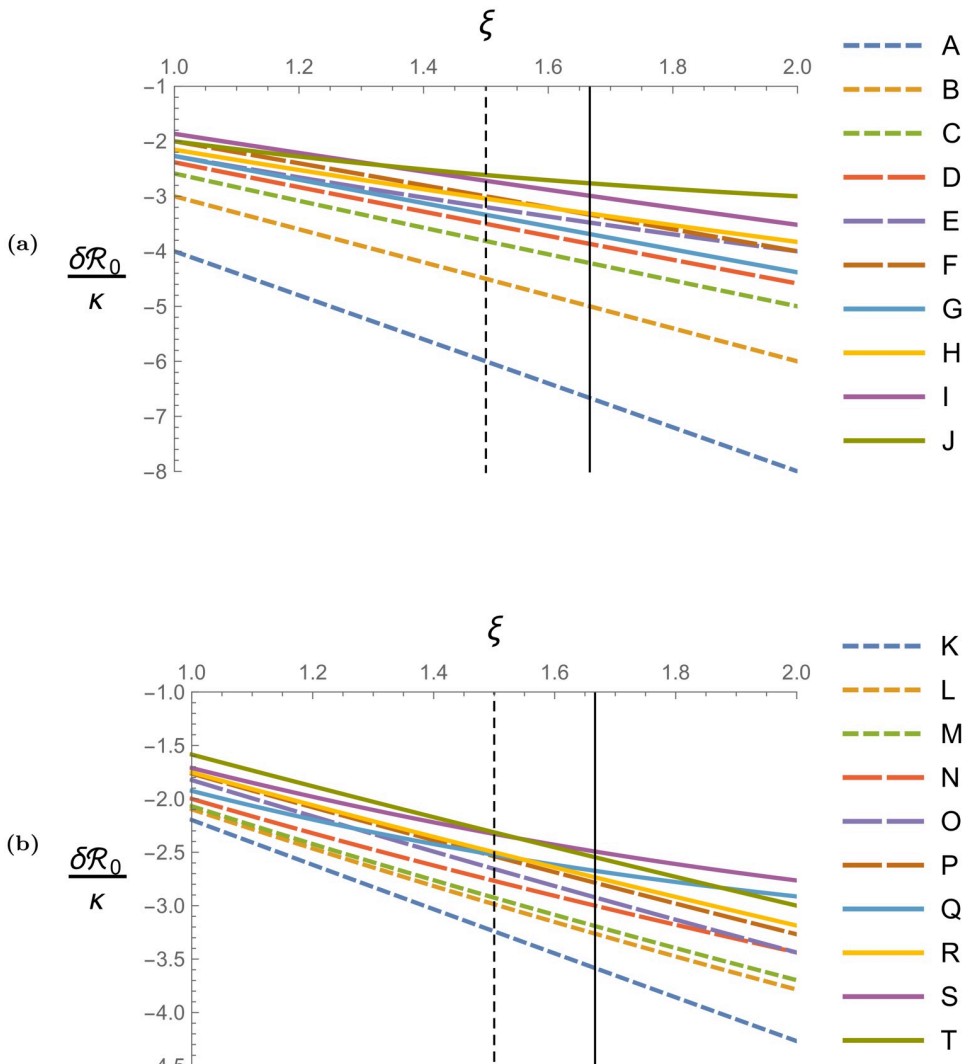

**Fig 5. Basic reproduction number perturbation (scaled by κ) as functions of ξ induced by the non-compliant blocks pictured in Fig 4.** In Fig. 5a, line A represents a single non-compliant site, line B represents a block of two non-compliant sites, line C represents a block of three non-compliant sites, lines D, E, and F represent different configurations of blocks of four non-compliant sites, and lines G, H, I, and J represent different blocks of five non-compliant sites. All lines in Fig. 5b represent different blocks of six non-compliant sites. Note the change in vertical scale between Figs. 5a and 5b. In both plots, the solid black vertical line $\xi = 5/3$ is the largest value of $\xi$ achievable when $p^{\nu} = \mu_0$, and corresponds to the case of no control in non-compliant sites and strongly efficacious control in compliant sites. The dotted vertical line $\xi = 1.5$ corresponds to the maximum possible value of $\xi$ in the limit of strongly efficacious non-compliant control $\mu_N/p^{\nu} \to \infty$.

Fig 4, and that larger and smaller values of $\xi$ produce distributions which are slightly more and less tightly clustered, respectively, around the sites with fewer border connections. The perturbations $\delta \mathcal{R}_0/\kappa$ and no hopping eigenvectors $\mathbf{I}^h_{(0)}$ and $\mathbf{I}^{\nu}_{(0)}$ in Figs 4 and 5 were calculated by solving Eqs (19) and (20) using the *eigensystem* function in *Mathematica 11.0*.

Fig 5 shows that $\delta \mathcal{R}_0$ generally decreases as $\xi$ increases, meaning that small mosquito hopping is most beneficial for suppressing outbreak potential when $\xi = 2$, and least beneficial when $\xi = 1$. When $\xi = 1$, for a given configuration size, $\delta \mathcal{R}_0/\kappa$ increases as the configuration border length decreases. For example, for the sites of size six in Fig 3, configuration T has the

smallest border (ten connections between the non-compliant block and the surrounding compliant sites), and gives the largest $\delta\mathcal{R}_0/\kappa$ of all the six site configurations when $\xi = 1$. The borders of configurations O, P, Q, R, and S have thirteen connections to surrounding compliant sites, and give the next largest six-site $\delta\mathcal{R}_0/\kappa$ values. The borders of configurations K, L, M, and N have fourteen connections to compliant sites, and give the smallest $\delta\mathcal{R}_0/\kappa$ values. For configurations which have the same number of sites and boundary lengths, the configurations with sites more tightly packed together tend to give larger $\delta\mathcal{R}_0/\kappa$ values when $\xi = 1$. As $\xi$ increases, Fig 5 shows that $\delta\mathcal{R}_0/\kappa$ tends to decrease at roughly similar rates for configurations with visually similar geometries, such as configurations C, D, and G, or configurations Q and S, for example. For configurations J, Q, and S, $\delta\mathcal{R}_0/\kappa$ decreases with $\xi$ more slowly and "least linearly" in comparison to the other configurations, and these configurations give the largest $\delta\mathcal{R}_0/\kappa$ values when $\xi = 2$. Configurations J, Q, and S are also the only configurations which have interior sites with no border connections.

It is important to note that first order perturbation results are invariant under "bending" symmetry operations which act on the non-compliant blocks and do not change their nearest neighbor configurations. For example, a straight line of $n$ non-compliant sites will give the same $\delta\mathcal{R}_0$ as an 'L,' 'S,' or 'U' shaped line of $n$ non-compliant sites, and the values of the non-zero components of $\mathbf{I}^h_{(0)}$ and $\mathbf{I}^v_{(0)}$ will be identical for all arrangements. For non-compliant blocks of sizes one site through six sites, the configurations pictured in Fig 4 generate all possible values of $\delta\mathcal{R}_0/\kappa$ for a given value of $\xi$, assuming that the blocks are not part of the neighborhood boundary. Any configuration of size one through six not pictured in Fig 4 can be obtained from one of the pictured configurations through a number of bending symmetry operations, and the corresponding value of $\delta\mathcal{R}_0/\kappa$ will be found in Fig 5.

## Optimized control

**No hopping and infinite hopping limits.**    Tables 3 and 4, and Fig 6 show optimized control results for the cases of no mosquito hopping, infinitely fast mosquito hopping, and homogeneous systems (at any hopping rate). These correspond to the special cases where we can find analytic expressions for $\mathcal{R}_0$ that do not depend on the spatial distribution of compliant

**Table 3. Optimal dollar per day costs in the no-hopping $\omega = 0$ limit as a function of the fraction of compliant sites, assuming model parameter values in Table 1 and control parameter values in S1 Appendices Table A.** For the special cases of 100% and 0% compliance, the above optimal application costs are equivalent to the optimal costs for a 100 site homogeneous neighborhood at any hopping rate.

| Fraction Compliant | Optimal Costs | | |
|---|---|---|---|
| | **Combined Controls** | **Aerial Spray Only** | **Door-to-Door Only** |
| 1 | $3.08 | $5.02 | $3.08 |
| less than 1 | $5.02 | $5.02 | $0.00 |

**Table 4. Optimal application frequencies in units of inverse days in the no-hopping $\omega = 0$ limit as a function of fraction of compliant sites, assuming model parameter values in Table 1 and control parameter values in S1 Appendices Table A.** For the special cases of 100% and 0% compliance, the above optimal application frequencies are equivalent to the optimal application frequencies for a 100 site homogeneous neighborhood at any hopping rate.

| Fraction Compliant | Optimal Application Frequency | | | |
|---|---|---|---|---|
| | **Aerial Spray: Combined Controls** | **Door-to-Door: Combined Controls** | **Aerial Spray Only** | **Door-to-Door Only** |
| 1 | 0.00 | 0.007279 | 0.2083 | 0.007279 |
| less than 1 | 0.2083 | 0.00 | 0.2083 | 0.00 |

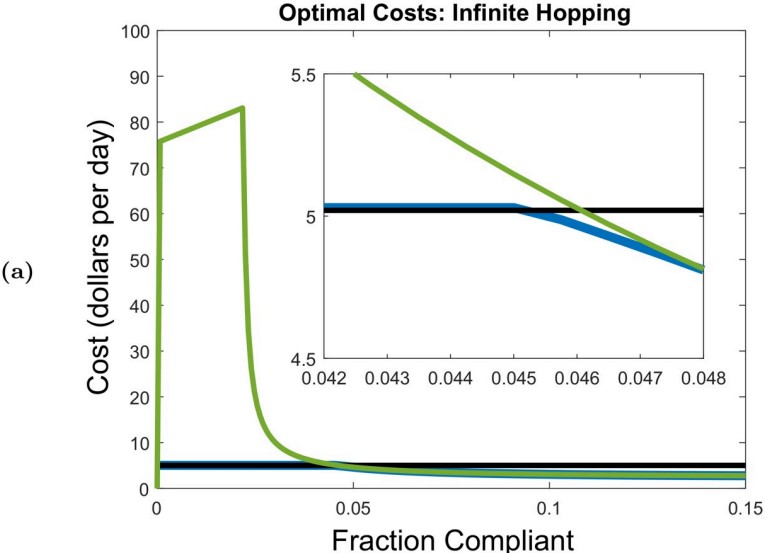

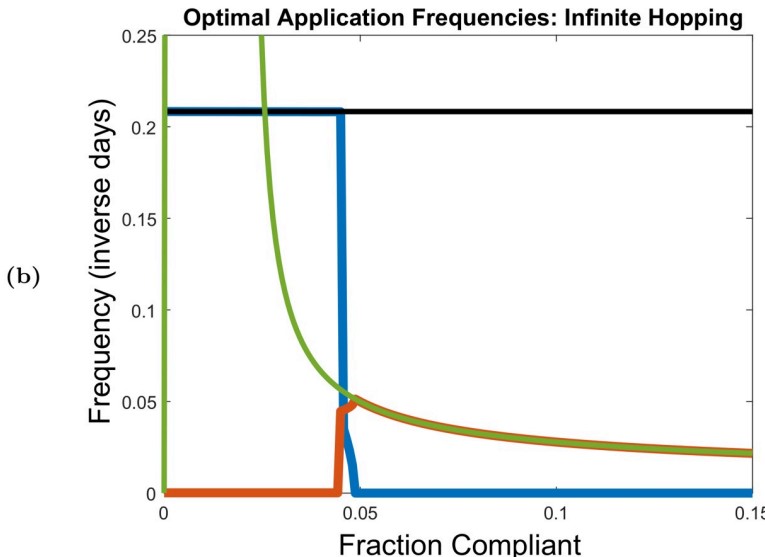

**Fig 6. Optimal costs and application frequencies in the infinite hopping $\omega \rightarrow \infty$ limit as functions of percent compliance, assuming model parameter values in Table 1 and control parameter values in S1 Appendices Table A.**
For the special cases of 100% and 0% compliance, the above optimal costs and application frequencies are equivalent to the optimal costs and application frequencies for a homogeneous neighborhood at any hopping rate (assuming a neighborhood comprised of 100 sites). In Fig. 6a, the blue curve is the optimal costs for combined aerial spray and door-to-door strategies, while the green and black curves are the optimal costs for door-to-door only control and aerial only control, respectively. The crossover points in the zoomed-in inset show the cut-off compliance levels, beyond which combined control strategies or door-to-door only control becomes more cost effective than aerial only control. In Fig. 6b, the blue and the red curves are the optimal aerial and door-to-door application frequencies, respectively, for combined control strategies, while the green and black curves are the optimal application frequencies for door-to-door only control and aerial only control, respectively. The optimal door-to-door only costs increase linearly between 0% and 2.1% compliance in Fig. 6a, and the optimal door-to-door only frequencies are equal to 1 day$^{-1}$ (not in the range of Fig. 6b).

houses. In the no hopping limit, Table 4 gives the cost-optimal application frequencies required to bring $\mathcal{R}_0$ to one as a function of percent compliance for combined aerial spray and door-to-door control, aerial spray alone, and door-to-door control alone. Table 3 gives the corresponding optimized control costs. We find that unless the system is 100% compliant, the optimal control action is to apply only aerial spray once every 4.80 days at a cost of $5.02 per day on average. This daily cost reflects the fact that we set the aerial spray costs to $24.10 per application. At 100% compliance, the optimal control action is to apply only door-to-door control once every 4.5 months at a cost of $3.08 per day on average. This cost corresponds to spending $422.90 to apply door-to-door control to the entire 100-house neighborhood once every 4.5 months. Finding that door-to-door control is never recommended below 100% compliance reflects the fact that at a hopping rate of zero, the basic reproduction number is determined solely by the non-compliant sites (if there are any in the neighborhood), together with the fact that door-to-door control has no influence on non-compliant sites at zero hopping rate. For homogeneous systems, $\mathcal{R}_0$ is independent of hopping rate, and Tables 3 and 4 therefore imply that for any $\omega$, the optimal control action for a 100% compliant system is door-to-door only control applied once every 4.5 months, and that the optimal control action for a 100% non-compliant system is aerial spray only control applied once every 4.80 days.

Fig 6 plots the optimal costs and application frequencies as a function of percent compliance in the infinitely fast hopping limit. At 0% compliance, Fig 6 is in agreement with the 0% compliance values in Tables 3 and 4. Fig 6 only shows between 0% and 15% compliance for visual clarity, but we note that when compliance approaches 100%, the infinitely fast hopping costs and application frequencies curves approach the 100% compliance values in Tables 3 and 4. From Fig 6, we see that below 4.58% compliance, the optimal control action is aerial spray only, applied once every 4.80 days. Above 4.80% compliance, the optimal control action is door-to-door control only, with optimal application frequencies decreasing as compliance increases. At 4.80% compliance, door-to-door control must be applied once every 19.5 days at a cost of $4.74 per day on average, which reflects a cost of $92.12 for a single door-to-door application applied to a 4.80% compliant neighborhood. The door-to-door only daily cost and application frequency reduce to $3.08 per day and once every 4.5 months, respectively, as percent compliance approaches 100%. We thus see that only within the narrow compliance interval (4.58%, 4.80%) does optimal control require combined aerial spray and door-to-door strategies. In this compliance interval, the combined controls are applied less frequently than they would be if they were used on their own, and optimal costs are lower than the aerial spray only and door-to-door only costs.

The flat black lines in Fig 6 indicate that compliance does not influence the optimized control costs and application frequencies for aerial spray alone in the infinite hopping limit (as is expected to be the case for any hopping rate). However, optimized application frequencies and control costs for door-to-door control alone do vary based on percent compliance in the infinite hopping limit. At 0% compliance, Fig 6 shows that the cost-optimal action for door-to-door only control is to never spend any money on control, which reflects the fact that door-to-door only control can not influence a system at 0% compliance. Between 0% and 2.18% compliance, Fig 6a shows that optimal costs for door-to-door control only increase linearly with percent compliance, and that the corresponding optimal application frequencies are found to be at the upper bound of once per day (note this is above the range shown in Fig 6b). This implies the system is uncontrollable under door-to-door control alone when compliance is below 2.18% in the infinite hopping limit. As compliance increases beyond 2.18%, the system becomes controllable under door-to-door control, and optimal door-to-door only costs and application frequencies rapidly reduce towards the 100% compliance values given in Tables 3 and 4. Note that although the system becomes controllable with door-to-door control at 2.18%

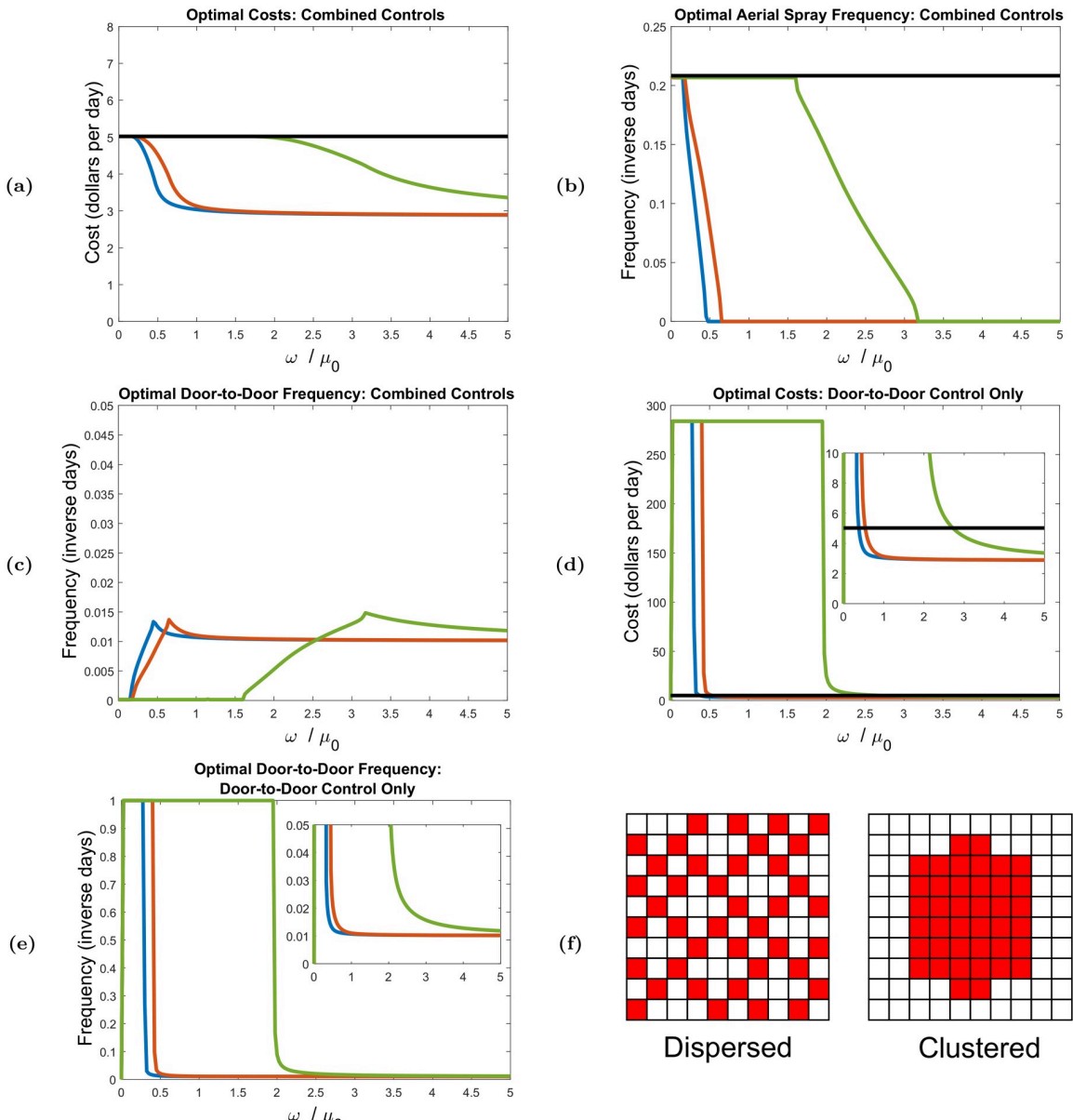

**Fig 7. Optimal controls and application frequencies for the 60% compliance dispersed and clustered distributions in Fig. 7f, where white squares indicate compliant sites, and red squares indicate non-compliant sites.** Blue and red curves correspond to the dispersed distribution under periodic and reflecting boundary conditions, respectively. The differences in costs and frequencies between reflecting and periodic boundary conditions is negligible for the clustered distributions, and the green curves in the figures correspond to the clustered distribution under either reflecting or periodic boundaries (these curves coincide). The black line in Figs. 7a, 7b and 7d corresponds to aerial spray control only (e.g. used without door-to-door control). The crossover points in the zoomed-in inset in Fig. 7d are the cut-off hopping rates beyond which door-to-door only control becomes more cost-effective than aerial spray only control.

compliance, Fig 6a shows that using door-to-door control alone does not become cheaper than aerial control alone until compliance levels surpass 4.65%.

**Finite hopping rates.** The effects of varied finite hopping rates are shown in Fig 7. Here, we show how optimized costs and application frequencies depend on hopping rate for specific clustered and dispersed 60% compliance distributions under both periodic and reflecting boundary conditions in a $10 \times 10$ neighborhood grid. Note that we plot these quantities against

$\omega/\mu_0$ rather than $\omega$. The quantity $\omega/\mu_0$ represents the average number of hops a vector makes over an average lifespan, which sets a scale for 'fast' and 'slow' hopping. Hopping is considered fast when $\omega/\mu_0 \gg 1$, and slow when $\omega/\mu_0 \ll 1$. We also consider clustered and dispersed 20% compliance distribution, and find results visually and qualitatively similar to those given in Fig 7. The corresponding 20% compliance plots are given in S1 Appendices Sec. 7.

We find that for a given compliance level, the different boundary conditions yield negligible differences in optimized cost and frequency curves for the clustered configurations, and yield qualitatively and numerically similar curves for the dispersed configurations. Generally, the periodic boundary systems are slightly cheaper to control than the corresponding reflecting boundary systems, and require slightly less frequent control applications. Fig 7 also shows that, unlike boundary conditions, compliance clustering can have a strong effect on optimal controls. Clustered systems are much more expensive to control than the corresponding dispersed systems, and they require much more frequent control applications. Further, Fig 7 shows that as hopping rates increase, optimized control becomes cheaper and needs to be applied less often, and that for small hopping rates, the optimal control action is to apply aerial only control once every 4.80 days. Focusing on any one neighborhood configuration and boundary condition, we see that as the hopping rate increases from zero, there exists a threshold hopping rate where the optimal control action transitions from an aerial spray only strategy to a combined aerial spray and door-to-door strategy. At even larger hopping rates, we find a second threshold where the optimal control action transitions from a combined control strategy to a door-to-door only control strategy. For the clustered 60% compliance distribution pictured in Fig 7f, under either boundary condition, optimal control action calls for combined strategies for hopping rates $\omega \in (1.65\mu_0, 3.18\mu_0)$. For hopping rates below and above this interval, optimal control action calls for aerial spray only and door-to-door control only, respectively. The corresponding intervals for the dispersed 60% compliance distribution are given by $(0.175\mu_0, 0.475\mu_0)$ under periodic boundary conditions and $(0.200\mu_0, 0.650\mu_0)$ under reflecting boundary conditions. The corresponding intervals for 20% compliance distributions are given S1 Appendices Sec. 7. Generally, at a lower level of compliance, the intervals are found to be wider and begin at larger values of $\omega$.

Fig 7 shows that for aerial spray alone, optimal control costs and application frequencies are independent of hopping rate, as expected from the fact that a system subject to only aerial control is homogeneous. Optimal results for door-to-door only control, however, are strongly dependent on hopping rate. For $\omega = 0$, the results in Fig 7 are consistent with the no-hopping results in Tables 3 and 4. Here, we see that the optimal action under door-to-door control alone is to never spend money on control. This reflects the fact that door-to-door control has no influence on the system's basic reproduction number at zero hopping unless compliance is at 100%. For each boundary condition and each compliance configuration pictured in Fig 7f, there exists a hopping rate interval where the system is controllable under door-to-door control alone, but is more expensive to control with door-to-door control than with aerial spray. For hopping rates below this interval (excluding $\omega = 0$), the system is uncontrollable with door-to-door control alone, and door-to-door only control is optimally applied as frequently as possible (once per day). Above this interval, the system is controllable under door-to-door only and is cheaper to control than with aerial spray only. The intervals are given by $(1.98\mu_0, 2.75\mu_0)$ for the clustered distribution under either boundary condition, $(0.300\mu_0, 0.400\mu_0)$ for the dispersed distribution under periodic boundary conditions, and $(0.425\mu_0, 0.550\mu_0)$ for the dispersed distribution under reflecting boundary conditions. The intervals for clustered and dispersed 20% compliance (see S1 Appendices Sec. 7) are generally found to be wider and begin at larger values of $\omega$.

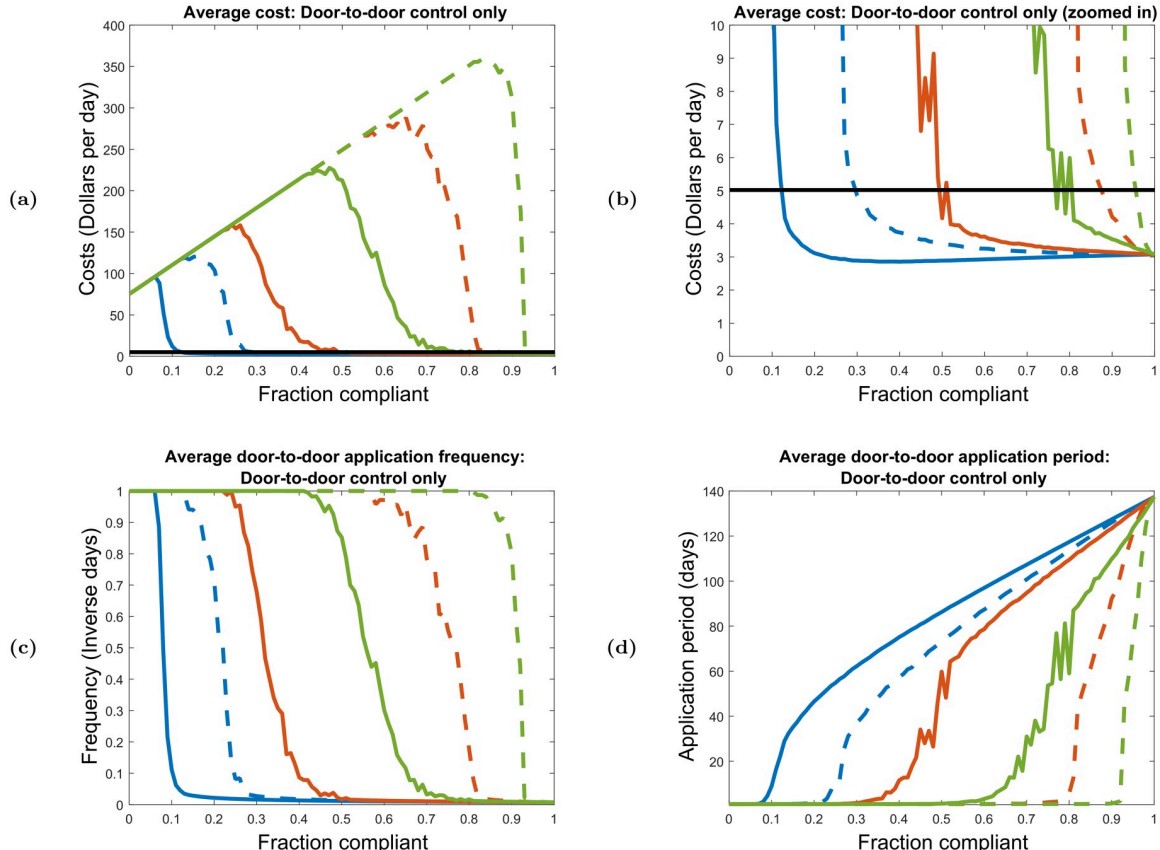

**Fig 8. Door-to-door only control costs, application frequencies, and application periods required to bring $\mathcal{R}_0$ to unity (or as low as possible when the system is uncontrollable) as a function of percent compliance.** Blue, red, and green curves correspond to the hopping rates $\omega = 5\mu_0$, $\omega = \mu_0$, and $\omega = 0.5\mu_0$, respectively. For each value of percent compliance and hopping rate, we average over 200 random neighborhood compliance configurations that are either highly clustered (dashed curves) or randomly dispersed (solid curves), assuming periodic boundary conditions. The crossover points in the zoomed-in plot in Fig 8b indicate cut-off compliance levels beyond which door-to-door only control becomes more cost-effective than aerial only control on average. Figures 8c and 8d represent the same information; we display both application period and application frequency for visual clarity.

Figs 8 and 9 show the results of our randomized door-to-door control only analysis. Generally, we see that for a given level of clustering, controllability increases with hopping rate, and that control costs and application frequencies decrease on average. Likewise, for a given hopping rate, the more dispersed distributions tend to be more controllable and cheaper to control than the clustered distribution. In Fig 8, we see that for a given hopping rate and level of clustering, there exists a region of low compliance where average costs increase linearly with percent compliance and the corresponding average application frequencies are equal to 1 day$^{-1}$. This compliance region corresponds to the region in Fig 9 where nearly all generated configurations for that hopping rate and clustering level are uncontrollable. As compliance levels increase beyond this region, average costs and average frequencies quickly drop as more and more of the generated configurations become controllable. For each hopping rate and clustering level in Fig 9, we note the existence of a compliance *controllability interval*. At compliance levels below the controllability interval, no generated configurations are controllable, and at compliance above the interval, all generated configurations are controllable. Within the interval, a non-zero fraction of the generated configurations are controllable. We also note the existence of a compliance *cost-effectiveness interval*. At compliance levels below the cost-

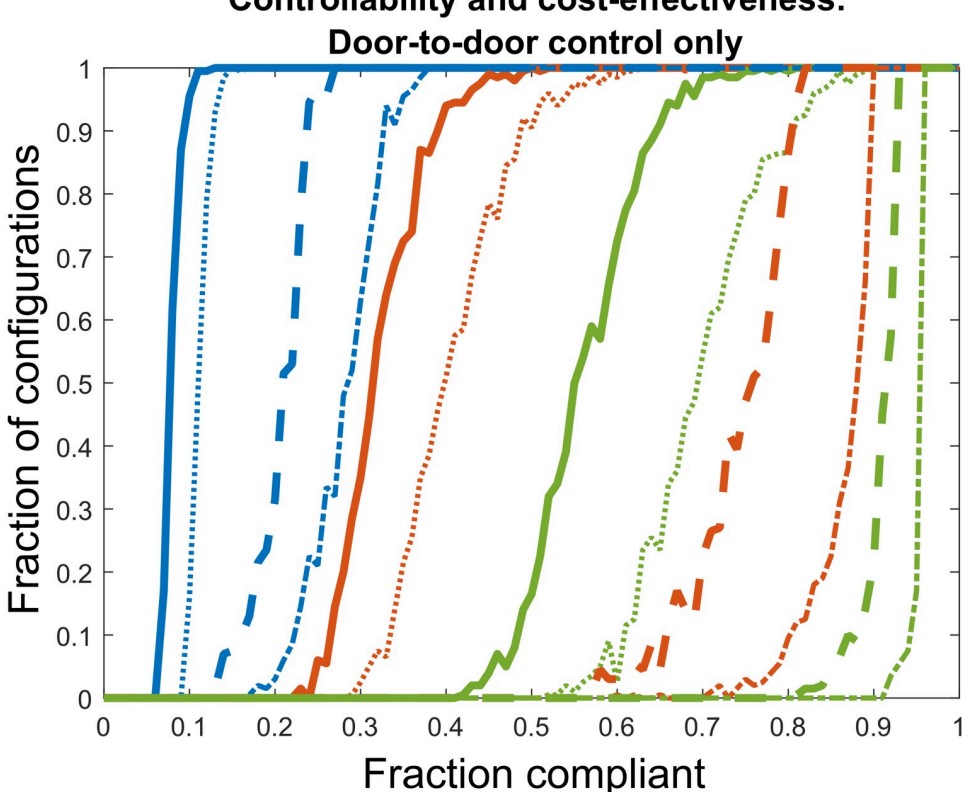

**Controllability and cost-effectiveness:**
**Door-to-door control only**

**Fig 9. Shown here are the fraction of neighborhood configurations that can be controlled with door-to-door alone as a function of percent compliance, as well as the fraction of configurations that are more cost-effective to be controlled with door-to-door control compared to aerial spraying.** Results are shown for three values of the vector hopping rate: Blue, red, and green curves correspond to the hopping rates $\omega = 5\mu_0$, $\omega = \mu_0$, and $\omega = 0.5\mu_0$, respectively. For each value of percent compliance and hopping rate, we average over 200 random neighborhood compliance configurations which are either highly clustered or randomly dispersed, assuming periodic boundary conditions. Solid curves represent the fractions of randomly dispersed configurations that are controllable with door-to-door control (e.g. those where $\mathcal{R}_0$ can be brought to unity), and the neighboring dotted curves represent the fractions of randomly dispersed configurations that are both controllable and cost-effective (e.g. those that are both controllable and cheaper to control with door-to-door control than with aerial spray). The dashed curves represent the fractions of highly clustered configurations that are controllable, and the neighboring dashed-dotted curves represent the fractions of highly clustered configurations that are both controllable and cost-effective.

effectiveness interval, no generated configurations are cheaper to control than with aerial spray alone, and at compliance levels above the interval, all generated configurations are cheaper to control than with aerial spray alone. These controllability and cost-effectiveness compliance intervals are given, respectively, by (7%, 12%) and (10%, 16%) for $\omega = 5.0\mu_0$ dispersed, (13%, 26%) and (17%, 37%) for $\omega = 5.0\mu_0$ clustered, (23%, 49%) and (29%, 62%) for $\omega = 1.0\mu_0$ dispersed, (56%, 81%) and (60%, 89%) for $\omega = 1.0\mu_0$ clustered, (42%, 81%) and (52%, 89%) for $\omega = 0.5\mu_0$ dispersed, and (79%, 92%) and (92%, 95%) for $\omega = 0.5\mu_0$ clustered.

## Discussion

### Summary of main results

Throughout this paper, we have shown how mosquito motion and control compliance influence $\mathcal{R}_0$, and determined cost-optimal strategies for suppressing outbreak potential with combinations of aerial spray and door-to-door control. These findings are interrelated; the more

abstract and theoretical $\mathcal{R}_0$ results provide the effective spatial distributions actually under control when outbreak potential is reduced, and these distributions in-turn provide visual intuition and insight into the mechanisms underlying the more biologically practical optimized control numerical results. These results, and the interrelations among them, are summarized here and discussed in detail in the following subsections.

We find that $\mathcal{R}_0$ will generally decrease with the vector hopping rate unless the system is spatially homogeneous, in which case hopping rate will have no effect on $\mathcal{R}_0$. This implies increased mosquito motion helps decrease the potential of epidemic outbreaks. We also find that $\mathcal{R}_0$ generally increases with the number and degree of clustering of non-compliant sites. Thus, having large connected blocks of non-compliant sites increases the potential for initially large and rapid epidemic outbreaks. When compliance is very low with moderate to slow hopping rates, when the hopping rate is very low, or when the non-compliant sites are highly clustered with moderate to slow hopping rates, we find that the system can not be controlled with door-to-door efforts alone, and adulticide aerial spray must be used in order to reduce $\mathcal{R}_0$ to unity. Under any of these conditions, door-to-door control is extremely ineffective on a system-wide level, and the optimal action is to control with aerial adulticide spray only. As these conditions are relaxed, so compliance or hopping rate increases, or non-compliant houses become less clustered, the optimal control action changes to a combined door-to-door and aerial spray strategy. For large hopping rates and large levels of compliance, door-to-door control is extremely effective on a system wide level, and the optimal control action calls for a door-to-door only strategy. In general, the efficacy of area-wide aerial spray control is independent of the rate of mosquito motion, compliance levels, and compliance clustering, while the efficacy of door-to-door control is highly dependent upon these factors. The results of this paper show that useful, actionable advice for cost-optimal preventative control can be obtained from a relatively simple linear optimization problem under the non-linear constraint $\mathcal{R}_0 = 1$. This control problem formulation is more accessible to disease managment planners without an advanced mathematical background as compared to typical optimal control theory techniques [39], which themselves are not guaranteed to yield direct practical control advice like optimized application frequencies.

## Controlling infectious spatial distributions

In this paper, we focus on controlling outbreak potential by reducing $\mathcal{R}_0$. This amounts to controlling outbreaks assuming that the infectious hosts and vectors are distributed such that they produce the largest possible asymptotic disease growth rate under disease dynamics linearized about the disease-free equilibrium. These "worst-case" spatial distributions are given by the eigenvectors $\mathbf{I}^v$ and $\mathbf{I}^h$ of the second generation matrices $\underline{\mathcal{M}}$ and $\mathcal{M}$, respectively, corresponding to the eigenvalue $\mathcal{R}_0^2$. The spatial distributions $\mathbf{I}^v$ and $\mathbf{I}^h$ provide a visual interpretation of what is actually being controlled when reducing $\mathcal{R}_0$, and may provide useful intuition for researchers and workers hoping to adapt our baseline results to specific conditions encountered in the field.

For the special case of a spatially homogeneous system, our analysis shows that $\mathbf{I}^v$ and $\mathbf{I}^h$ are uniformly distributed over the entire neighborhood, regardless of the vector hopping rate $\omega$. For spatially inhomogeneous systems, $\mathbf{I}^v$ and $\mathbf{I}^h$ generally depend on $\omega$, but are found to be uniformly distributed in the infinitely fast hopping limit $\omega \to \infty$, regardless of spatial structure. Thus, in order to reduce $\mathcal{R}_0$ for either infinite hopping or homogeneous systems, effectively, one must control an infectious vector population that is spread uniformly over the entire neighborhood.

For the case of infinitely slow hopping (meaning no mosquito motion between sites), the mathematical analysis at $\omega = 0$ shows that $\mathbf{I}^v$ and $\mathbf{I}^h$ will be non-zero only within the neighborhood's non-compliant sites. However, the manner in which the distributions vary throughout the non-compliant sites is left indeterminate due to the degeneracy in the eigenspaces of $\underline{\mathcal{M}}$ and $\mathcal{M}$ for the eigenvalue $\mathcal{R}_0^2$ when $\omega = 0$. In other words, the no-hopping mathematical analysis alone cannot identify which infectious distributions over the neighborhood's non-compliant sites are in any sense more biologically meaningful or more relevant to designing control strategies for reducing $\mathcal{R}_0$ when $\omega = 0$. Fortunately, first order perturbation analysis resolves this degeneracy, identifying particular eigenvectors $\mathbf{I}_{(0)}^v$ and $\mathbf{I}_{(0)}^h$ from the $\mathcal{R}_0^2$ eigenspaces of $\underline{\mathcal{M}}$ and $\mathcal{M}$ at $\omega = 0$. Specifically, for small but non-zero hopping rates, first order perturbation analysis shows that the eigenvectors $\mathbf{I}^v$ and $\mathbf{I}^h$ vary continuously with $\omega$ and reduce to the particular no-hopping eigenvectors $\mathbf{I}_{(0)}^v$ and $\mathbf{I}_{(0)}^h$ as $\omega$ approaches zero. The results of our perturbation analysis, particularly those displayed in Figs 4 and 5, show that $\mathbf{I}_{(0)}^v$ and $\mathbf{I}_{(0)}^h$ will generally be distributed in the larger, more clustered blocks of connected non-compliant sites, and are more heavily weighted towards the interiors of non-compliant blocks away from the boundaries in connection with the surrounding compliant sites. Here, the precise meaning of "more clustered" depends on the relative difference between compliant and non-compliant control efficacy as quantified by the unitless parameter $\xi \in [1, 2]$. When compliant and non-compliant control efficacies are nearly equal in strength, $\xi \approx 1$, and a "more clustered" block of a given block size is taken to mean a block with a smaller boundary. The parameter $\xi$ increases with compliant control efficacy and decreases with non-compliant control efficacy, and when $\xi = 2$, a "more clustered" block of a given size is taken to mean a block with more interior sites not directly connected to the block's boundary. Values of $\xi$ between 1 and 2 represent a type of weighting between the two notions of clustering. Note that for large non-compliant blocks where boundaries are small in comparison to total block size, the distinction between the two notions of clustering will be inconsequential. When $\omega$ is small but non-zero (meaning that the majority of vectors do not leave their emergence site, and those that do will most likely make only one hop over the course of their lifetime), the no-hopping eigenvectors are perturbed, and the resulting slow hopping eigenvectors spread into the compliant sites directly surrounding the larger, more clustered non-compliant blocks. From these results, we conclude that in order to reduce $\mathcal{R}_0$ for zero and small non-zero hopping rates, effectively, one must control an infectious vector population that is distributed in and directly adjacent to the largest, most clustered, non-compliant blocks.

First order perturbation theory lifts some, but not all of the eigenvector degeneracy present in the no-hopping limit. For example, if there exist two largest, most tightly clustered non-compliant blocks in a neighborhood with identical spatial structures of compliant and non-compliant connections, first order perturbation analysis determines that $\mathbf{I}_{(0)}^h$ and $\mathbf{I}_{(0)}^v$ will be distributed entirely within the two blocks, but will yield no information on how the distributions should be split or shared between the two blocks. Further, first order perturbation results are invariant under "bending" symmetry operations which do not change the nearest neighbor configurations of a non-compliant block, and this symmetry may cause some degeneracy to linger at small non-zero hopping rates. For example, if there exists a straight line and a bent 'L', 'U', or 'S' shaped line of non-compliant sites of identical length, first order perturbation theory cannot determine how $\mathbf{I}_{(0)}^h$ and $\mathbf{I}_{(0)}^v$ will be split among the two lines when they are the only two non-compliant blocks in a neighborhood. These lingering degeneracies could be further lifted by retaining terms of higher order in $\omega/\mu_0$ and performing a higher order perturbation analysis, but the mathematics are more involved and are outside the scope of this work.

Higher order terms are able to break bending symmetry, and can be expected to, for example, recognize the bent 'L' shaped non-compliant line as more clustered and more capable of producing an epidemic outbreak than a straight line of the same length. The corresponding refined no-hopping eigenvectors will presumably depend not only on the intraclustering within individual non-compliant blocks, but also the interclustering between groups of non-compliant blocks, and will be approximated by the first order perturbation no-hopping eigenvectors discussed in this paper. Further, one can show that an $n^{th}$ order perturbation expansion will generally yield perturbed slow hopping eigenvectors that are non-zero in compliant sites at most $n$ hops out and away from the non-compliant blocks occupied in the no-hopping limit. These higher order effects represent small corrections to the first order effects, and will become more and more appreciable as $\omega/\mu_0$ becomes larger (up to $\omega/\mu_0 = 1$ where the series expansions of $\underline{\mathcal{M}}$ and $\mathcal{M}$ diverge). Taking these observations together with our mathematical results for no hopping, slow hopping, and infinitely fast hopping, we conclude that in order to reduce $\mathcal{R}_0$ for any vector hopping rate, effectively, one must control an infectious vector population that is concentrated at and distributed around the larger, more clustered non-compliant blocks, such that the degree of spreading away from the larger, more clustered non-compliant blocks is a non-decreasing function of $\omega$.

## Mosquito motion and disease controllability

One of the primary results of the $\mathcal{R}_0$ analysis developed in this paper is that mosquito motion is generally beneficial to disease controllability. In particular, we show that an infinitely fast hopping system is easier to control than a slow hopping system (e.g. the infinitely fast $\mathcal{R}_0$ is less than or equal to the finitely slow hopping $\mathcal{R}_0$), and that a slow hopping system is easier to control than a no-hopping system. For these special cases where analytic expressions for $\mathcal{R}_0$ can be derived, the possible values of the system-wide basic reproduction number are found to be bounded below by the single site compliant basic reproduction number $\mathcal{R}_{0C}$ and bounded above by the single site non-compliant basic reproduction number $\mathcal{R}_{0N}$. Specifically, when $\omega = 0$, $\mathcal{R}_0$ is equivalent to $\mathcal{R}_{0N}$ when the system is less than 100% compliant, and is equivalent to $\mathcal{R}_{0C}$ when the system is 100% compliant. For small non-zero values of $\omega$, first order perturbation analysis shows that $\mathcal{R}_0$ decreases by a small amount relative to $\mathcal{R}_{0N}$ (unless the system is either 100% compliant or non-compliant). The magnitude of this decrease is proportional to the relative hopping rate $\omega/\mu_0$ and the parameter $\kappa$, where $\kappa$ decreases as a function of the relative non-compliant control efficacy $\mu_N/\mu_0$. When $\omega \to \infty$, $\mathcal{R}_0$ has the same mathematical form as the single-site basic reproduction numbers, but with the death and emergence rates averaged over the entire neighborhood, meaning that $\mathcal{R}_0$ smoothly decreases from $\mathcal{R}_{0N}$ to $\mathcal{R}_{0C}$ as compliance increases from 0% to 100%.

The decrease in $\mathcal{R}_0$ from no hopping to slow hopping to infinite hopping can be understood intuitively by visualizing the effective infectious vector distribution that must be controlled. At zero hopping, the effective distribution lies entirely in non-compliant blocks and is therefore only influenced by $\mathcal{R}_{0N}$, while at slow hopping, the effective distribution spreads by a small amount from the larger, more clustered non-compliant blocks into the surrounding compliant sites, and therefore becomes less influenced by $\mathcal{R}_{0N}$ while becoming more influenced by $\mathcal{R}_{0C}$. This results in an overall decrease in the system-wide $\mathcal{R}_0$. At infinite hopping, the effective distribution is equally influenced by all sites as it spreads uniformly over the entire neighborhood and, consequently, the system-wide $\mathcal{R}_0$ is weighted between $\mathcal{R}_{0N}$ and $\mathcal{R}_{0C}$ according to the fraction of compliant and non-compliant sites.

The general $\omega$-dependent behavior of the effective mosquito distributions that must be controlled in order to reduce the basic reproduction number provides insight into the general $\omega$-

dependent behavior of $\mathcal{R}_0$ outside of the analytically tractable special cases. This distribution is concentrated in the larger, more clustered non-compliant blocks at $\omega = 0$, and spreads away from these blocks into the other neighborhood sites as $\omega$ increases, eventually becoming uniformly distributed as $\omega \to \infty$. Importantly, when $\omega$ increases, the magnitude of the distribution in the non-compliant blocks occupied at $\omega = 0$ will never increase. This follows from the fact that the total number of vectors represented by the distribution is constant, and so an increase in the magnitude of the distribution at one site must come at the expense of a decrease in the magnitude of the distribution at another site. Therefore, when an increase in $\omega$ causes the distribution to spread and increase in a compliant site, one can expect a decrease in the system-wide $\mathcal{R}_0$ due to an increased influence of $\mathcal{R}_{0C}$ and decreased influence of $\mathcal{R}_{0N}$. If an increase in $\omega$ causes the distribution to spread and increase in a non-compliant site, the amount of increase ultimately comes at the expense of the distribution levels in the larger, more clustered non-compliant blocks occupied at $\omega = 0$, meaning that the influences of $\mathcal{R}_{0C}$ and $\mathcal{R}_{0N}$ on the distribution will be unchanged, and so one can expect a non-increase in the system-wide $\mathcal{R}_0$. We thus conclude that, in general, $\mathcal{R}_0$ will be a non-increasing function of hopping rate for any compliance spatial structure, and that mosquito motion is generally beneficial for controlling and preventing epidemic outbreaks. In essence, slow mosquito motion allows infectious vector populations to remain concentrated in and around non-compliant neighborhood sites where their populations can best flourish, and these conditions can produce very large localized outbreaks which are difficult to control. Fast mosquito motion, on the other hand, prohibits infectious vector populations from remaining localized in sites ideal for rapid disease growth, and so the worst-case scenario disease growth rates indicated by $\mathcal{R}_0$ will generally be smaller for more rapid mosquito motion. The non-increase in $\mathcal{R}_0$ as a function of hopping rate is corroborated indirectly by the optimized control results in Fig 7 for the specifically considered 60% compliance distributions (and also by Fig. B in S1 Appendices Sec. 7 for 20% compliance distributions). Here, we see that optimal control costs are non-increasing functions of hopping rate (aside from the special case of $\omega = 0$ under door-to-door control only, which optimally employs no control actions at no cost), thus indicating that the control effort required to bring $\mathcal{R}_0$ to unity is a non-increasing function of hopping rate.

In addition to showing the general non-increase of $\mathcal{R}_0$ with hopping rate, our results imply that the degree to which $\mathcal{R}_0$ decreases with hopping rate is determined by how close the system is to homogeneous. Specifically, when compliant and non-compliant control efficacies are more similar, an increased hopping rate will yield smaller reductions in $\mathcal{R}_0$ and will therefore be less beneficial for disease control. When the system is exactly homogeneous, meaning equivalent compliant and non-compliant control efficacies (which occurs under aerial spray only control or under any control scheme in a 100% compliant or non-compliant system), $\mathcal{R}_0$ is independent of hopping rate, so an increase in hopping rate will have no effect on disease controllability. At infinitely fast hopping, Fig 6a shows that when compliant and non-compliant control efficacies are most similar, percent compliance has a minimal effect in decreasing $\mathcal{R}_0$. Intuitively, the decrease in $\mathcal{R}_0$ at large hopping rates is driven by the effective infectious distribution's spread from the "bad" non-compliant sites into the "good" compliant sites, so when the distinction between "bad" and "good" is negligible, the number of "good" sites that the distribution is able to reach is largely inconsequential. The dependence of the $\omega$-dependent decrease in $\mathcal{R}_0$ on spatial homogeneity is illustrated directly by our first order perturbation analysis, specifically in Fig 3. First order perturbation analysis shows that the rate at which $\mathcal{R}_0$ decreases with hopping increases with the unitless parameter $\xi \in [1, 2]$, where smaller values of $\xi$ indicate more similar non-compliant and compliant control efficacies. Values of $\xi \approx 1$ represent cases where the difference between compliant and non-compliant control efficacy is

negligible, while the value $\xi = 2$ represents the case of strongly and weakly efficacious compliant and non-compliant control, respectively, with a negligible extrinsic incubation period $1/p^v \to 0$. Non-negligible incubation periods yield values of $\xi$ which decrease from values smaller than 2 towards the value 1 as compliant and non-compliant control efficacy become more similar.

Interestingly, our first order perturbation results indicate that mosquito motion is especially beneficial for control when the extrinsic incubation period is negligibly small. This effect arises due to the fact that the effective infectious vector distribution being controlled is most concentrated within blocks of non-compliant sites. When the incubation period is non-negligible, newly infected vectors within a non-compliant block have an opportunity to die inside that block before they become infectious and propagate the disease, despite the fact that non-compliant control efficacy may be small to non-existent, and so the stronger control efficacy in the surrounding compliant sites will have a diminished effect in reducing overall disease levels. Stated another way, for a non-negligible incubation period, if a number of newly infected (but not infectious) vectors leave their non-compliant blocks of origin and die due to strong control in the surrounding compliant sites, a non-zero fraction of them would have died naturally before becoming infectious in the absence of strong compliant control, and so strong compliant control acting on this fraction will not have provided any actual benefit for suppressing disease growth. On the other hand, when the incubation period is negligibly small or non-existent, all newly infected vectors within a non-compliant block become instantly infectious and immediately begin propagating the disease. If a number of these infectious vectors were to move into compliant sites and die quickly due to strong compliant control, the fraction that would have died naturally before becoming infectious in the absence of the strong compliant control is exactly zero, and so strong compliant control acting on the entire group will provide benefits for suppressing disease growth.

It is important to note that the conclusions presented here regarding the influence of hopping rate on $\mathcal{R}_0$ depend crucially on the manner in which we define the effects of control in compliant and non-compliant sites. When vectors spread from "bad" (bad in the sense of a greater single site basic reproduction number) non-compliant sites into "good" compliant sites, they are traveling from regions of high emergence rate and low death rate into regions with low emergence rate and high death rate. This feature is the ultimate source of the observed non-increase of $\mathcal{R}_0$ with hopping rate. If, on the other hand, the "bad" sites were defined to have high emergence rates and high death rates while the "good" sites were defined to have have low emergence rates and low death rates, increased hopping rates have the potential to increase $\mathcal{R}_0$ by allowing the large numbers of infected vectors born in the "bad" sites to escape into the "good" sites where they can live longer and find more opportunities to spread disease. This situation will never arise in the context of compliant and non-compliant control unless there exist natural heterogeneities that are independent of control.

### Practical control advice: Door-to-door vs. area-wide aerial spray

The cost-optimal door-to-door and area-wide aerial spray application frequencies required for reducing $\mathcal{R}_0$ to unity are influenced by the rate of mosquito motion, the level of door-to-door compliance, as well as the degree of compliance clustering. The manner in which these factors influence the optimal choice between door-to-door control only, aerial spray control only, and a combined integrated vector management strategy is a central result of this work. At low levels of compliance, door-to-door control alone can not control the system unless the mosquito hopping rate is sufficiently large. Mathematically, this effect arises from the fact that the system-wide $\mathcal{R}_0$ is determined by an infectious vector population which is distributed in and

around the larger, more clustered non-compliant blocks of sites. In this sense, door-to-door control can only reduce the system-wide $\mathcal{R}_0$ when the infectious vectors originating in the non-compliant blocks have sufficient mobility such that they spend enough time in compliant sites to feel the effects of strong compliant control. This explains why a greater level of clustering of non-compliant sites is more difficult to control than a dispersed distribution of non-compliant sites; vectors existing in an extremely deep block of non-compliance require an extreme amount of mobility in order to feel the effects of door-to-door control in compliant sites. However, even at infinitely fast mosquito motion, under our model and control parameters, the system is uncontrollable under door-to-door control at compliance levels below about 2%, regardless of compliance clustering. Simply put, even if the vector population is able to be essentially eliminated in compliant sites, such sites must comprise about 2% of the total area, at minimum, in order to have any hope of door-to-door control alone preventing an epidemic outbreak. More realistically, at finite (potentially small) hopping rates, much greater levels of compliance are required for door-to-door controllability, especially if the compliance distribution is highly clustered.

When a system is uncontrollable with door-to-door control alone, control efforts must be supplemented by area-wide spraying, and the combined action of door-to-door control and aerial spraying can potentially be more cost-effective than either strategy used alone. For a given distribution of compliant sites, when hopping rates are far too slow for the system to be controllable with door-to-door control alone, the optimal action is to use only area-wide spraying. In such cases, an inability to spray frequently due to societal concerns, budgetary constraints, or resistance concerns is detrimental to disease control, and no amount of door-to-door control can make up for the deficit. As hopping rates approach the door-to-door controllability threshold from below, the optimal control action becomes to supplement aerial spraying with door-to-door control. Here, the disease is still uncontrollable under door-to-door only, so an inability to conduct frequent area-wide spraying is still detrimental to outbreak prevention. However, mosquito motion is fast enough such that a non-trivial fraction of infectious vectors originating in non-compliant sites will travel to compliant sites where they experience the effects of door-to-door control. As hopping rates continue to increase past the door-to-door controllability threshold, aerial spray is optimally applied less frequently, and door-to-door is optimally applied more frequently. At these hopping rates, an inability to conduct aerial spraying will not be detrimental to outbreak prevention, but will require sub-optimal spending on door-to-door efforts in order to control the system. At large enough hopping rates, the system will not only be controllable under door-to-door efforts alone, but will also be much more cost effective under door-to-door alone than under aerial spray alone. Here, the optimal control action is to apply only door-to-door control.

Generally speaking, systems with lower levels of compliance and greater levels of clustering will have larger intervals of hopping rates where optimal control actions call for using aerial spray, either alone or combined with door-to-door efforts. Likewise, systems with smaller hopping rates will have greater numbers of compliance levels and randomly dispersed distributions of compliant sites for which the the optimal control action is to apply aerial spray only in comparison to systems with faster hopping rates. For a given hopping rate, there will be a greater number of highly clustered compliance distributions for which the optimal control action is only aerial spray in comparison to more randomly dispersed compliance distributions. As shown in Fig 9, the effects of compliance clustering is diminished at larger hopping rates. This follows from the notion that highly mobile vectors experience the effects of compliant and non-compliant sites in a more averaged sense in correspondence to the fraction compliance in the neighborhood, where the actual spatial distributions of compliance and non-compliance become increasingly irrelevant as $\omega \to \infty$.

It is important to note than even when a system is controllable under door-to-door control alone, using only door-to-door control may not be more cost-effective than aerial spray alone, despite door-to-door's higher control strength and lasting effects in compliant sites relative to aerial spray. Specifically, Fig 9 indicates that for a given hopping rate, the number of compliance configurations which are controllable under door-to-door alone is smaller than the number of configurations which are cost-effective under door-to-door alone. The differences between the number of configurations which are controllable and the number of configurations which are cost-effective, however, tends to be small outside of the narrow compliance ranges where systems transition from always uncontrollable to always controllable. On the other hand, Fig 7 shows that, for a given compliance configuration, the range of hopping rates over which the system is controllable is greater than the range of hopping rates which are cost-effective, and that the differences between the two ranges are rather small. In any event, we obtain the following 'rules of thumb'—when a system is in the range of uncontrollable to almost controllable, the optimal control action is to apply only aerial spray; when a system is in the range of almost controllable to controllable and slightly more cost-effective, the optimal control action is to apply a combined aerial spray door-to-door strategy; when a system is controllable and in the range of slightly more to much more cost-effective, the optimal control action is to apply door-to-door control only.

## Concluding remarks

We have shown that mosquito motion, door-to-door control compliance levels, and spatial clustering of compliant sites play an important role in determining whether or not vector-borne disease can be controlled by area-wide aerial spraying and/or door-to-door control, as well as the most cost-effective strategies for control, in a neighborhood scale system. We find that, in general, increased mosquito motion, increased compliance levels, and decreased compliance clustering are all beneficial for the efficacy of door-to-door control efforts. Interestingly, using a modeling framework similar to ours, Lutambi et al. [17] have previously shown that clustered control can be slightly beneficial for reducing mosquito populations under low compliance levels of residual adulticide and larval control strategies. By contrast, our model shows that clustered door-to-door control is never beneficial for reducing outbreak potential. It is unclear as to whether this difference is due to a difference in control goals (outbreak suppression versus population suppression), or due to differences in model structure.

We note the numerical results presented here are all based on a disease-related parameter set representative of typical values associated with vector-borne diseases such as Zika or dengue in North America. Our intent here is not to provide specific control advice which can be responsibly applied directly in the field. Rather, we have focused on providing mechanistic insight into the biological factors which should, in conjunction with additional practical considerations that can not be reliably modeled, be considered when designing a real-world integrated vector management strategy. Our specific numerical results are most appropriate for use as a simplified baseline example case from which one can build some mathematical intuition for control efficacy on neighborhood scales.

The optimization scheme for reducing $\mathcal{R}_0$ to unity presented in this paper is, to the best of our knowledge, the first example of optimized $\mathcal{R}_0$ control where results can be consistently and directly interpreted in terms of application frequencies for real-world control schemes that are directly connected to real-world experimentally measurable control parameters. While the numerical results presented here represent a simplified baseline example, the optimization scheme serves as a template which can be easily adapted by workers and researchers seeking results more fine-tuned to system-specific details. The essential mathematical ingredients for

calculating $\mathcal{R}_0$ (e.g. the functions relating control application frequencies to changes in uncontrolled model parameter values and the mathematical forms of the second generation matrices) provided here will remain unchanged under modified control parameters, cost parameters, disease parameters, and spatially heterogeneous natural death rates, emergence rates, and host densities. With these tools, one needs only a basic familiarity with built-in local optimization packages found in mathematical software (such as the *fmincon* function in *Matlab*) in order to find optimized control frequencies.

The model presented in this paper, while somewhat generic, is limited in several aspects, and it will be interesting in future work to consider how adding more realistic detail will modify our baseline results. Of greatest interest will be to consider spatially heterogeneous natural emergence rates, death rates, and host density in order to study the interaction between compliance clustering and natural host and mosquito clustering. Such modifications, however, will require no changes in model structure. Larval classes and host motion, on the other hand, are potentially important factors omitted from our model whose inclusion would increase model complexity. The addition of a larval class with a larval death rate would allow modelling of area-wide larvicide as a control strategy, and we expect that the additional area-wide spray efficacy given by larvicides can diminish the roles of mosquito motion, compliance levels, and compliance clustering on disease controllability. The effects of adding host motion to our model are difficult to predict, as human motion throughout a neighborhood is not straightforward to include in an ODE compartmental model. In contrast to mosquito motion, a random walk analogous to a free Brownian particle is a poor description of human motion in a neighborhood. This is because humans tend to spend most of their time in and around their home site. A better description of host motion would be that of a Brownian harmonic oscillator centered at the host's home site. In an ODE compartmental model, such a description would translate to a host population where a given host's presence is distributed in a roughly Gaussian manner about their home site when in equilibrium. Additionally, one could include random host hops into sites not directly connected to home sites in order to simulate hosts who frequent the homes of friends and family living in distant locations throughout the neighborhood. If the effects of host motion are indeed important factors, we expect models of host motion to dominate the dynamics of disease spread when the mosquito hopping rate is very small. By building further biological detail on top of the base model and optimization scheme presented here, powerful tools for designing real-world control schemes tuned towards system-specific details can be further developed.

## Supporting information

**S1 Appendices. Mathematical details and additional results.** This series of appendices gives mathematical expressions for the Laplacian matrices under periodic and reflecting boundary conditions (Sec. 1), gives an expression for the equilibrium vector population distribution (Sec. 2), derives mathematical expressions for the second generation matrices leading to the $\mathcal{R}_0$ eigenvalue problem (Sec. 3), provides detailed derivations of the $\mathcal{R}_0$ expressions in simplified limiting cases (Sec. 4), provides details related to the perturbation theory formalism (Sec. 5), explains the mathematical relationships between controlled model parameters and real-world control strategies' application frequencies (Sec. 6), provides additional optimized control results for specific 20% compliance distributions (Sec. 7), and shows the relationship between total epidemic size and $\mathcal{R}_0$ for aerial spray and door-to-door control strategies (Sec. 8).
(PDF)

**S1 Data and code. Optimized control code and data sets.** This compressed file contains the optimized control data presented in this paper, as well as the Matlab programs used to generate the data.
(ZIP)

## Author Contributions

**Conceptualization:** Kevin A. Caillouët, Suzanne L. Robertson.

**Methodology:** Jeffery Demers, Sharon Bewick, Folashade Agusto, Kevin A. Caillouët, Suzanne L. Robertson.

**Supervision:** William F. Fagan.

**Writing – original draft:** Jeffery Demers.

**Writing – review & editing:** Jeffery Demers, Sharon Bewick, Folashade Agusto, Kevin A. Caillouët, William F. Fagan, Suzanne L. Robertson.

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
