## [Decision Letter · Decision Letter 0]

31 Jan 2020

Dear Dr Demers,

Thank you very much for submitting your manuscript 'Managing disease outbreaks: The importance of vector mobility and spatially heterogeneous control' for review by PLOS Computational Biology. Your manuscript has been fully evaluated by the PLOS Computational Biology editorial team and in this case also by independent peer reviewers. The reviewers appreciated the attention to an important problem, but raised some substantial concerns about the manuscript as it currently stands. While your manuscript cannot be accepted in its present form, we are willing to consider a revised version in which the issues raised by the reviewers have been adequately addressed. We cannot, of course, promise publication at that time.

Your revisions should address the specific points made by each reviewer.

In particular, I agree with the reviewers that while the paper derives several interesting insights, it is in its current form too long, too inaccessible, and too much focused on technical aspects. The revised version should therefore fundamentally streamline the manuscript, shift technical parts into a supplementary material and place much more emphasis on the biological/epidemiological utility of the results. 

Please return the revised version within the next 60 days. If you anticipate any delay in its return, we ask that you let us know the expected resubmission date by email at ploscompbiol@plos.org. Revised manuscripts received beyond 60 days may require evaluation and peer review similar to that applied to newly submitted manuscripts.

Sincerely,

Roger Dimitri Kouyos

Associate Editor

PLOS Computational Biology

Rob De Boer

Deputy Editor

PLOS Computational Biology

[LINK]

Reviewer's Responses to Questions

**Comments to the Authors:**

Reviewer #1: Authors conducted a study on “Managing disease outbreaks: The importance of vector mobility and spatially heterogeneous control.” Their approach seems novel and exciting in mathematical epidemiology. Also, their study can provide a critical/useful tool for cost effectiveness analysis of vector-borne infectious diseases. However, some critical points from mathematical and epidemiological aspects should be considered for publication in PLOS Computational Biology.

Major comments

It is insufficient for the validity of controlling only R0 (the basic reproduction number less than unity) in their spatial model. In general, the basic reproduction number for a single-patch mathematical model is the key indicator whether outbreaks will occur or not. However, it is not clearly shown whether it would be the case for multi-patch models.

For example, multi-patch models for vector-borne diseases with human movement showed disease persistence even though R0 is less than 1 in each patch (JTB 2009, The Effects of Human Movement on the Persistence of Vector- Borne Diseases). The results highlight human movement can cause the disease persistence even when R0 < 1.

Authors should provide extensive analysis on the final size relation (the total number of infected and the basic reproduction number) and furthermore, the effects of vector mobility on the final size relation should be discussed as well.

Therefore, the authors should carefully validate if it is sufficient to control only the basic reproduction number.

Reviewer #2: This paper presents a mathematical model aimed at answering a set of very interesting questions about optimization of mosquito control methods and their impact on mosquito-borne disease. Overall, this work is very interesting, timely, and useful; however, I worry that the paper in its current form is far too dense and without a focused direction, and I found it difficult to extract major points from this work. I have included a few comments below that I hope will be of aid to the authors.

1. In Methods II, the authors walk carefully through the process of how the basic reproductive number was obtained. This section was perhaps the most dense of the methods section and it read as material more appropriate for a theoretical biology or applied mathematics journal. Would it be possible to break this section down into key points and leave details to the appendices? If this approach was taken, the authors could focus on the R0 expressions obtained (such as those in equation 23) and explain the importance of each. The authors could then follow up with the hopping scenario-specific discussions of R0.

2. Related to the hopping scenarios: It would be interesting to get the authors' perspectives on how each of these scenarios relates to different mosquitoes and mosquito borne disease. It seems like finitely slow hopping is the only realistic scenario of the three. These scenarios are mathematically interesting, but are they biologically relevant?

3. I think interpretation of the results would benefit from a table of parameter descriptions in the main text. The table could include only those parameters that are important to reading and understanding figures.

4. It would also be worthwhile to have more clear reasoning behind the choice of metrics in figures. For instance, why do the authors present the metric on the vertical axis in Fig 6? What is the value of looking at omega/mu_0 in Figs 8-9? Are there other metrics that would perhaps be easier to interpret biologically for these results?

5. What are they key results here? There were a number of interesting figures presented in the results; however it was difficult to parse the major points from the work as it is presented.

6. I think this work is very interesting and useful overall, but I worry that there is too much information presented here, which distracts from the intended message of the paper.

**Have all data underlying the figures and results presented in the manuscript been provided?**

Reviewer #1: Yes

Reviewer #2: Yes

PLOS authors have the option to publish the peer review history of their article (what does this mean?). If published, this will include your full peer review and any attached files.

Reviewer #1: No

Reviewer #2: No

---

## [Decision Letter · Decision Letter 1]

5 Jun 2020

Dear Dr. Demers,

Thank you very much for submitting your manuscript "Managing disease outbreaks: The importance of vector mobility and spatially heterogeneous control" for consideration at PLOS Computational Biology. As with all papers reviewed by the journal, your manuscript was reviewed by members of the editorial board and by several independent reviewers. The reviewers appreciated the attention to an important topic. Based on the reviews, we are likely to accept this manuscript for publication, providing that you modify the manuscript according to the review recommendations.

Sincerely,

Roger Dimitri Kouyos

Associate Editor

PLOS Computational Biology

Rob De Boer

Deputy Editor

PLOS Computational Biology

[LINK]

Reviewer's Responses to Questions

**Comments to the Authors:**

Reviewer #1: The authors have improved the paper; the paper is well written, with clear structure and careful explanations throughout. However, the authors have not attempted to show a relationship between the basic reproduction number, R0 and the final epidemic size (the one issue raised in the previous comments). The authors still need to carry out at least one simple case, which supports optimizing R0 is equally good as optimizing the final epidemic size.

**Have all data underlying the figures and results presented in the manuscript been provided?**

Reviewer #1: Yes

PLOS authors have the option to publish the peer review history of their article (what does this mean?). If published, this will include your full peer review and any attached files.

Reviewer #1: No
---

## [Editor Report · Decision Letter 2]

9 Jul 2020

Dear Dr. Demers,

We are pleased to inform you that your manuscript 'Managing disease outbreaks: The importance of vector mobility and spatially heterogeneous control' has been provisionally accepted for publication in PLOS Computational Biology.

Best regards,

Roger Dimitri Kouyos

Associate Editor

PLOS Computational Biology

Rob De Boer

Deputy Editor

PLOS Computational Biology

---

## [Editor Report · Acceptance letter]

13 Aug 2020

PCOMPBIOL-D-19-01657R2 

Managing disease outbreaks: The importance of vector mobility and spatially heterogeneous control

Dear Dr Demers,

I am pleased to inform you that your manuscript has been formally accepted for publication in PLOS Computational Biology. Your manuscript is now with our production department and you will be notified of the publication date in due course.

With kind regards,

Matt Lyles
